# Blood-based quantification of Aβ oligomers indicates impaired clearance from brain in *ApoE* ε4 positive subjects
Lara Blömeke [1,2], Fabian Rehn [1,2], Marlene Pils [1,2], Victoria Kraemer-Schulien [1], Anneliese Cousin[1], Janine Kutzsche [1], Tuyen Bujnicki [1], Silka D. Freiesleben [3,4], Luisa-Sophie Schneider [3,4], Lukas Preis[3,4], Josef Priller[3,5,6,7], Eike J. Spruth[3,5], Slawek Altenstein[3,5], Anja Schneider [8,9], Klaus Fliessbach[8,9], Jens Wiltfang [10,11,12], Niels Hansen [11], Ayda Rostamzadeh[13], Emrah Düzel[14,15], Wenzel Glanz[14], Enise I. Incesoy [14,15,16], Katharina Buerger[17,18], Daniel Janowitz[18], Michael Ewers[17,18], Robert Perneczky[17,19,20,21], Boris-Stephan Rauchmann[19,22,23], Stefan Teipel[24,25], Ingo Kilimann[24,25], Christoph Laske[26,27], Matthias H. Munk[26,28], Annika Spottke[8,9,29], Nina Roy[8], Michael T. Heneka [30], Frederic Brosseron [8], Michael Wagner [8,9], Sandra Roeske[8], Alfredo Ramirez [8,9,31,32,33], Matthias Schmid[8,34], Frank Jessen[8,11,31], Oliver Bannach[1,2], Oliver Peters[3,4] & Dieter Willbold [1,2,35] ✉

## Abstract

**Background** Quantification of Amyloid beta (Aβ) oligomers in plasma enables early diagnosis of Alzheimer's Disease (AD) and improves our understanding of underlying pathologies. However, quantification necessitates an extremely sensitive and selective technology because of very low Aβ oligomer concentrations and possible interference from matrix components.
**Methods** In this report, we developed and validated a surface-based fluorescence distribution analysis (sFIDA) assay for quantification of Aβ oligomers in plasma.
**Results** The blood-based sFIDA assay delivers a sensitivity of 1.8 fM, an inter- and intra-assay variation below 20% for oligomer calibration standards and no interference with matrix components. Quantification of Aβ oligomers in 359 plasma samples from the DELCODE cohort reveals lower oligomer concentrations in subjective cognitive decline and AD patients than healthy Control participants.
**Conclusions** Correlation analysis between CSF and plasma oligomer concentrations indicates an impaired clearance of Aβ oligomers that is dependent on the *ApoE* ε4 status.

## Plain language summary

People with Alzheimer's disease have difficulties with reasoning and communication. In Alzheimer's disease, small proteins called amyloid beta (Aβ) stick together, forming tiny clusters in the brain that eventually grow larger. In this study, we aimed to measure these clusters in the blood. When we tested our method on blood samples from 359 people, we surprisingly found that people with Alzheimer's disease and memory problems had fewer clusters of Aβ compared to healthy individuals. Our finding suggests that genetic factors may influence the body's ability to clear these clusters from the brain.

Although Alzheimer's Disease (AD) is the most prominent neurodegenerative disorder affecting 50 million people worldwide[1], there remains a lack of therapeutic options and diagnostic tools, such as a blood-based test suitable for use in primary care. As increasing evidence supports the role of Amyloid beta (Aβ) oligomers as the most toxic component in AD progression[2–4], these oligomers represent a promising biomarker candidate for diagnosis of AD and drug development. The level of Aβ oligomers in the brains of AD patients are higher and, because of the direct connection to brain parenchyma and liquor[5], also in cerebrospinal fluid (CSF), as supported by previous studies[6–9]. Moreover, more than 50% of monomeric brain Aβ is transferred and cleared in the periphery[10], reaching the blood via

blood-brain-barrier (BBB), blood-CSF barrier (BCSFB) or perivascular and glymphatic clearance[11].

Although the exact clearance mechanisms of Aβ oligomers from the brain and CSF to plasma remain largely unknown, earlier studies have confirmed the presence of Aβ oligomer species in plasma samples[12].

However, disease progression may lead to a reduction in Aβ oligomers in plasma samples because of their deposition in amyloid plaques and impaired clearance from the brain into the blood stream[13]. For example, an inverse correlation between efficiency of glymphatic clearance and oligomer size has been described[2]. Additionally, transport of Aβ across the BBB is affected in AD patients, especially in carriers of the AD risk gene allele

apolipoprotein E (*ApoE*) ε4[2,14]. Quantification of Aβ oligomer concentrations in plasma samples, especially in early disease stages, and in-depth analysis of dependencies between Aβ oligomers and different biomarkers will improve our understanding of the role of amyloid pathology in AD.

Previous studies have reported higher oligomer concentrations in AD patients[13,15–17]. All of the methods applied in these studies detect specific subtypes of Aβ oligomers, depending on the respective antibody used. For Aβ oligomers derived from the brain, a broad range of species was described, ranging from small molecular weight oligomers like dimers and trimers via 56mers and spherical oligomers like Aβ derived diffusible ligands (ADDLs) to high-molecular weight oligomers and protofibrils[18,19]. For these species, differences have been claimed for their neurotoxicity and pathologic mechanisms, like impairment of mitochondrial dysfunction, $Ca^{2+}$ homeostasis dysregulation and induction of tau pathology[18].

The most widely described method for detection of Aβ oligomers in plasma is the multimer detection system (MDS) which uses $Aβ_{1-42}$ to amplify the signal and is therefore a tool to measure the oligomerization tendency of on-pathway oligomers. Using this method, AD patients showed significantly increased MDS signal compared to the control group[15,20,21] and the correlation with cognitive decline using neuropsychological tests like MMSE and CERAD[20,22,23]. Other methods used oligomer specific antibodies to quantify those oligomer species that are recognized by the respective specific antibody[12,17,24]. A third method claimed to quantify the alpha-sheet content of oligomers in plasma using specifically designed alpha-sheet peptides[25].

In contrast to these methods, the surface-based fluorescence intensity distribution analysis (sFIDA) technology aims to quantify all oligomer species, irrespective of their conformation, morphology and size, all of them potentially relevant for disease development and progression. sFIDA is a versatile platform for quantification of protein aggregates in biofluids that features single particle sensitivity due to a microscopy-based readout and selectivity for aggregated Aβ because of the use of antibodies with overlapping or even identical linear epitopes at the N-terminus of Aβ (principle of sFIDA in Fig. 1a). Quantifying the total amount of oligomeric species is crucial for quantitation of target engagement in the development of anti-oligomeric drugs. New therapies aim to eliminate Aβ oligomers. Using a diagnostic tool that captures all oligomer species may show the effect of this anti-oligomeric drug irrespective of the exact mechanism of action and the target oligomer species.

As calibration standard for oligomer-based assay, we previously established protein conjugated silica nanoparticles (SiNaPs)[26–28]. Additionally, we demonstrated that sFIDA sensitively and specifically detects alpha synuclein (αSyn), Tau and Aβ oligomers in CSF samples[9,29]. Nonetheless, the reliable quantification of Aβ oligomers in plasma samples poses an even greater challenge, as plasma typically contains a 200-fold higher total protein

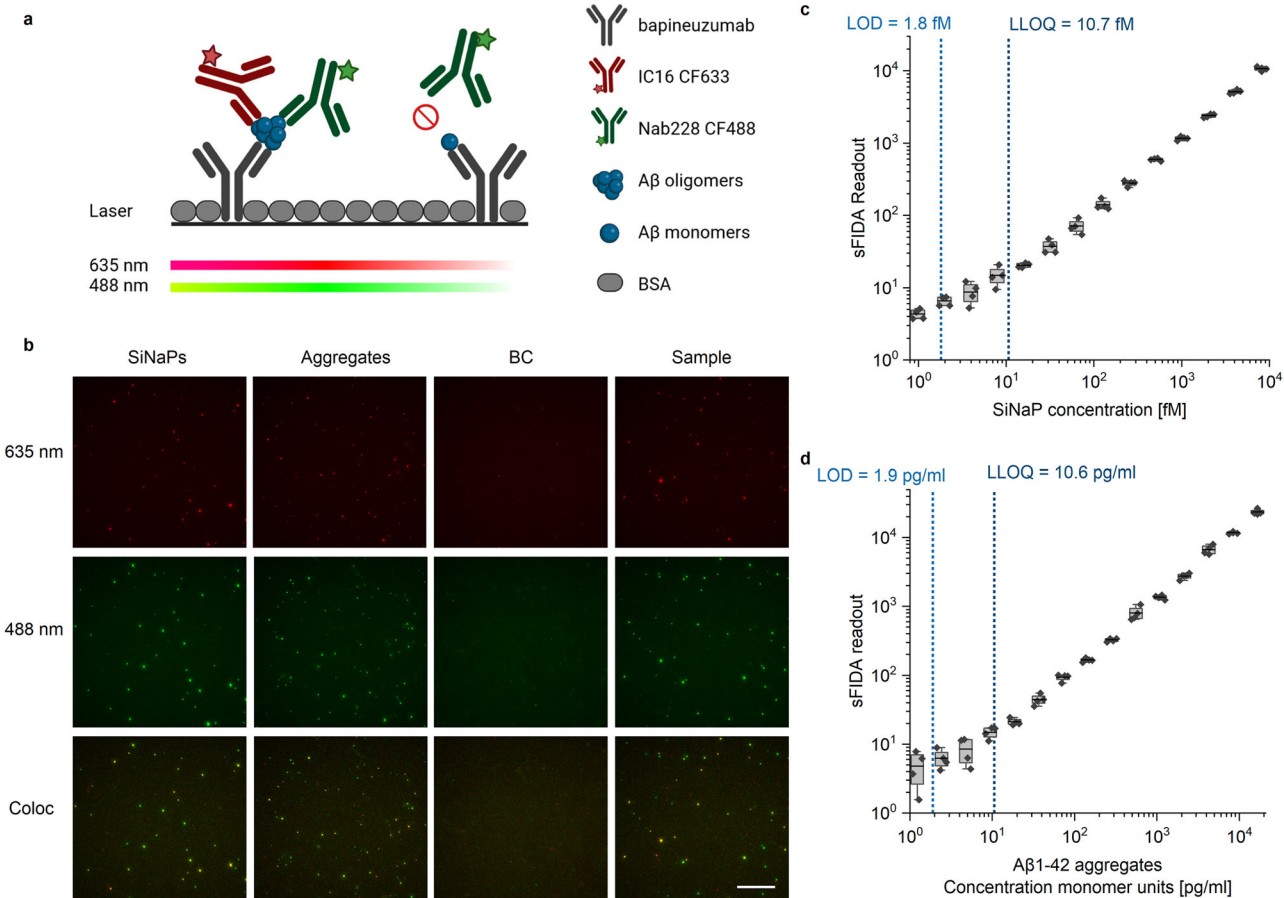

**Fig. 1 | Principle of sFIDA setup, imaging and calibration. a** The biochemical principle of sFIDA is similar to a sandwich ELISA with capture and detection antibodies directed against overlapping epitopes of the Aβ N-terminus. Therefore, monomers can be captured but not detected as the epitope is already occupied. After preparation, the assay surface is imaged using dual colour fluorescence microscopy (635 and 488 nm, respectively). *Created with biorender.com.* **b** Exemplary images of 500 fM SiNaPs coated with $Aβ_{1-15}$, aggregates composed of 564 pg/ml $Aβ_{1-42}$, a blank plasma (blank control, BC) and an AD plasma sample for the red (illumination with 635 nm) and green (illumination with 488 nm) fluorescence channels and colocalization. For imaging, the gray-scale value of 14-bit images was adjusted to min and max values of 750 and 7500, respectively. The scale bar is 50 μm. **c** Calibration curve of 1 fM to 8 pM $Aβ_{1-15}$ SiNaPs for the colocalization. **d** Dilution series of $Aβ_{1-42}$ aggregates consisting of 1.1 to 18,060 pg/ml $Aβ_{1-42}$ monomers. Boxplots include 25-50% intervals with a line for the mean. Whiskers present 1.5x the interquartile range. Limit of detection (LOD) and lower limit of quantification (LLOQ) were calculated as BC with a single- or ten-fold standard deviation. Standard deviations were calculated across the four replicates. Please note the logarithmic scale.

concentration than CSF[5]. This high protein background can lead to false negative readouts because of epitope masking, or false positive readouts because of interferences with human anti-mouse-antibodies[30]. Moreover, Aβ oligomer concentrations are expected to be in the low femto- to even atto-molar range[31], thus requiring an extremely sensitive method for detection.

In this report, an sFIDA assay for quantification of Aβ oligomers in plasma samples was developed and validated as a basic research project. We intended to quantitate the total Aβ oligomer levels in plasma samples of the DELCODE cohort to investigate the development of their concentrations during disease progression and their dependency from the *ApoE* ε4 status of the donors.

The sFIDA technology applied for quantifying Aβ oligomers in plasma samples demonstrated femtomolar sensitivity with negligible interference from matrix components making it suitable for quantitation of Aβ oligomers in plasma. Plasma samples from SCD and AD patients exhibited significantly lower Aβ oligomer concentrations compared to controls, particularly in amyloid-positive subjects. Correlation analyses between Aβ oligomers in CSF and plasma revealed a relationship between the two body compartments, influenced by disease progression, the presence of amyloid pathology, and the ApoE ε4 status of the donors.

## Methods
### SiNaPs
We used our previously developed silica nanoparticle standard (SiNaPs) as an assay control and for calibration. SiNaPs are small, spherical particles with a diameter of ~18.5 nm, which are functionalized with amino acids 1–15 of the Aβ peptide. Synthesis and characterization of the particles have been described previously[28,29] (Supplementary Fig. 1). The silica core of the particles was synthesized via the Stöber process and subsequently modified with APTES (3-aminopropyl(triethoxysilane), Sigma-Aldrich, St. Louis, USA) to create an aminated surface. As a crosslinker between protein and aminated particles, we used maleimido hexanoic acid (MIHA, abcr, Karlsruhe, Germany), which was activated using EDC (1-ethyl-3-(3-dimethylaminopropyl)carbodiimide, Sigma-Aldrich) and NHS (N-hydroxysuccinimide, Sigma-Aldrich). After washing, $Aβ_{1-15}$ (Peptides and Elephants, Henningsdorf, Germany) functionalized with cysteamine at the C-terminus was reacted with the maleimide group of SiNaPs to form a covalent attachment. $Aβ_{1-15}$ was added to achieve a theoretical protein load of ~18 $Aβ_{1-15}$ peptides per SiNaP. After 1 h, TCEP (Tris-(2-carboxyethyl)-phosphine, abcr, Karlsruhe, Germany) was added to prevent oxidation of the cysteamine-functionalized protein, and 1 h later, the reaction was terminated by adding 2-mercaptoethanol. SiNaPs were washed twice with $ddH_2O$. Prior to use, SiNaPs were subjected to ultrasonication for 15 s at 50% amplitude with a 1 s pulse - 1 s pause cycle.

### Aggregates
**Aβ aggregates.** $Aβ_{1-42}$ was purchased from Bachem AG (Bubendorf, Switzerland). 50 μg aliquots of $Aβ_{1-42}$ were dissolved in 1,1,1,3,3,3-hexafluoroisopropanol (HFIP, Sigma-Aldrich) and divided in 5 μg aliquots. HFIP was then evaporated in a vacuum concentrator (Vacufuge concentrator, Eppendorf, Hamburg, Germany) and stored at RT. 5 μg of $Aβ_{1-42}$ was dissolved in 5 μl dimethyl sulfoxide (DMSO, Sigma-Aldrich) for 10 min with shaking at 650 rpm (Thermomixer, Eppendorf). PBS and 1% sodium azide (AppliChem, Darmstadt, Germany) were added to achieve a concentration of 10 μM $Aβ_{1-42}$ containing 0.04% sodium azide. $Aβ_{1-42}$ was allowed to aggregate for 16 h at RT with shaking at 650 rpm. Aggregates were used directly or stored at −80 °C in 5 μl aliquots. Aβ aggregates have previously been characterized in Pils et al. using Thioflavin T assay (THT) and atomic force microscopy (AFM). Aggregates showed a monodisperse size distribution with a mean diameter of 2.7 nm[32]. We focused on $Aβ_{1-42}$ to prepare artificial aggregates, because in our hands their preparation is more robust and reproducible, and, because the capture and detection antibodies used here do not discriminate between $Aβ_{1-40}$ and $Aβ_{1-42}$.

**αSyn aggregates.** αSyn (expressed and purified in-house) was dissolved to 1 mg/ml in 20 mM Tris-HCl containing 100 mM sodium chloride (pH 7.4) and incubated for seven days at 37 °C with shaking at 1000 rpm. Aggregates were then sonicated for 60 s in 15 s intervals with a 1 s sonication pulse and a 1 s pause. Aggregates were aliquoted and stored at −80 °C. Preparation of α-Syn aggregates was based on Lohmann et al.[33]. For characterization, AFM measurements were used[33].

**Tau aggregates.** Full-length Tau (expressed and purified in-house) was dissolved in TBS buffer (Serva, Duisburg, Germany) containing 10-fold excess TCEP. Tau was centrifuged at 18,213 *g* for 1 h at RT and the Tau concentration in the supernatant was determined using UV-Vis spectroscopy. For aggregation, 8 μM heparin and 0.05% sodium azide were added to 15 μM Tau. The mixture was incubated at 37 °C with shaking at 300 rpm every 10 min for 10 days. Tau aggregates were characterized previously including AFM measurement and THT[34].

### Fluorescent antibodies
Fluorescently labelled detection antibodies were used for detection of SiNaPs and aggregates. IC16 (Heinrich-Heine-Universität Düsseldorf) was labelled with a red-fluorescent dye (CF633, Sigma-Aldrich), whereas Nab228 (Sigma-Aldrich) was labelled with a green-fluorescent dye (CF488, Sigma-Aldrich). The labelling process, the determination of concentration and degree of labelling are described in the manufacturer's protocol. Purification was carried out with a polyacrylamide bead suspension (Bio-Gel P-30, Bio-Rad Laboratories, Hercules, USA).

### Assay setup
Greiner BioOne 384 well plates (Kremsmünster, Austria) were used to ensure an adequate sample and replicate number within one assay. The biochemical principle of sFIDA was described previously[27]. The time course of the sFIDA workflow was illustrated in Fig. 2. For capturing, 40 μl of the humanized monoclonal antibody bapineuzumab (ProteoGenix, Schiltigheim, France) was used at a concentration of 0.625 μg/ml in 0.1 M carbonate buffer (Carl Roth, Karlsruhe, Germany) overnight at 4°C. Wells were washed using an automated microplate washer (405 LS Microplate Washer, BioTec, VT, USA), with five washing cycles with 80 μl TBS-T (TBS (Serva) containing 0.05% Tween (AppliChem)) and five washing cycles with 80 μl TBS. Washing with TBS-T and TBS was performed after each incubation step. After washing, the remaining binding sites were blocking using 0.5% BSA (AppliChem) in TBS containing 0.03% ProClin (Sigma-Aldrich) for 1.5 h at RT. After washing, we first applied 15 μl of LowCross buffer strong (Candor Bioscience, Wangen, Germany) to the wells to reduce matrix effects and then added 15 μl sample or SiNaPs and aggregates spiked in plasma. Samples were centrifuged at 2500 *g* for 5 min prior to analysis. The supernatant was transferred to a new tube and incubated on the plate for 2 h at RT. Fluorescently labelled IC16 CF633 and Nab228 CF488, each at 0.625 μg/ml in TBS containing 0.03% ProClin were first diluted in 0.1% BSA and 0.05% Tween and then centrifuged for 1 h at 100'000 xg. For detection, 20 μl per well of the probes were incubated for 1 h at RT. The TBS buffer was exchanged with 80 μl TBS-ProClin prior to measurement to prevent bacterial growth during measurement and storage. Calibration standards (SiNaPs and aggregates) were spiked and analysed in plasma to prevent plasma matrix effects on the calibration of the results.

**Influence of monomeric $Aβ_{1-40}$ and $Aβ_{1-42}$.** $Aβ_{1-40}$ was dissolved at 0.1 μg/ml in HFIP, whereas $Aβ_{1-42}$ was dissolved to 10 μM (approximately 45 μg/ml) and shaken for 24 h at RT and 600 rpm. Prior to analysis, monomeric Aβ was diluted to 25 nM in LowCross buffer strong and then to 100 pM in plasma.

**Influence of heterophilic anti-mouse antibodies (HAMA).** HAMA interference was analysed by spiking different concentrations of goat anti-mouse antibody (Thermo Fisher Scientific, Waltham, USA) in neat plasma. In addition to the capture antibody bapineuzumab, the Nab228 antibody was coated at 2.5 μg/ml in 0.1 M carbonate buffer to the glass surface to compare their HAMA interference in sFIDA.

## sFIDA workflow

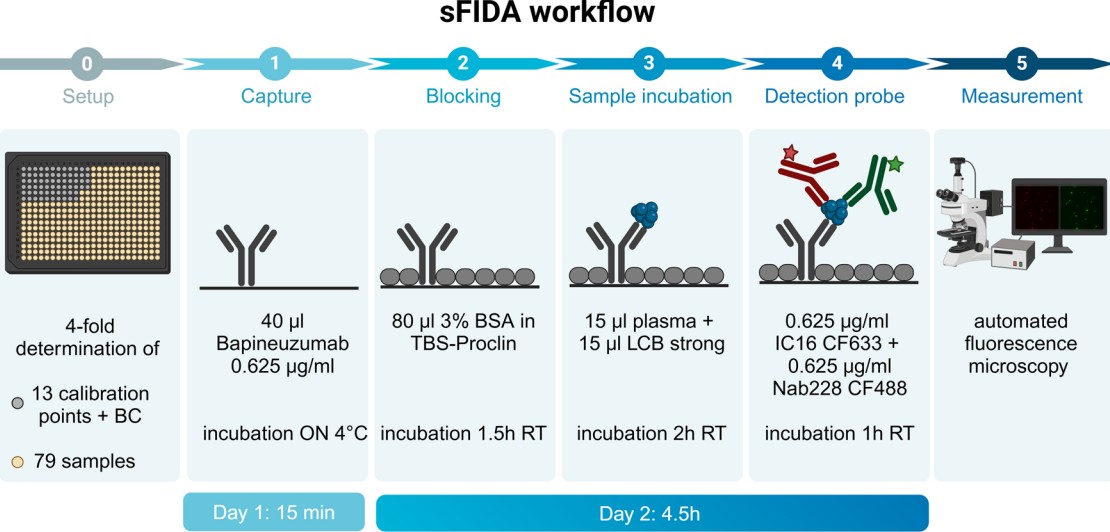

**Fig. 2 | Time course of the sFIDA workflow.** The use of 384 well plates allow a close-meshed concentration series and the determination of 79 patient samples on one plate in 4-fold replicate determination. The individual steps consist of an over night (ON) incubation of the capture antibody at 4 °C, a 1.5 h blocking step following by 2 h incubation of the plasma samples and 1 h incubation of detection antibodies. The final measurement is conducted by an automated fluorescence microscope. *Created with BioRender.com.*

**Influence of haemolysis**. Non-haemolytic plasma was spiked with different concentrations of haemolytic plasma to examine the effects of red blood cell haemolysis on assay results. Haemolytic plasma was prepared by freezing whole blood at −80 °C for 24 h and centrifugation at 1200 *g* for 15 min after thawing. Plasma prepared using this procedure is referred to as 100% haemolytic.

### Measurement
Measurement of the sFIDA assay surface was performed using total internal reflection microscopy (TIRF-M, Leica DMI6000B, Wetzlar, Germany) with 100x magnification, as described previously (excitation: 635 nm, emission filter: 705 nm; excitation: 488 nm, emission filter: 525 nm; exposure time both channels: 1000 ms; gain 1300)[35]. Each image consisted of 1,000,000 pixels (1000 × 1000 pixels) and in total, 100 images per sample (25 images per well) were measured, which covers 3.14% of the total well surface.

For measurement and calibration of the DELCODE cohort plasma samples, a third fluorescence channel was added (excitation: 405 nm, emission filter: 450 nm) for automated detection and elimination of artificial pixels, as described in statistics.

### sFIDAta
The in-house software tool sFIDAta was used for analysis of the images. sFIDAta enables automated detection and elimination of artefact-containing images and counting of pixels above a cutoff value. The cutoff value is defined as the grey-scale value, at which a predefined number of pixels in the blank control are counted and is determined individually for each fluorescence channel and for each experiment to compensate for fluorescence fluctuations. For analysis, a cutoff value of 0.05% (blank control exceeds 500 pixel) was chosen. PixelCount refers to the average number of pixels in an image above the cutoff value for each fluorescent channel, whereas the sFIDA readout describes the number of colocalized pixels that exceed the cutoff value in both fluorescence channels. Min-max filtering was applied to prevent possible remaining artificial images influencing outcomes after artefact detection. Min-max filtering excluded 10% of images per well with the highest fluorescence and 10% of images per well with the lowest PixelCounts[29].

The measurement procedure was extended by an additional step for analysis of DELCODE cohort samples and for inter-assay measurements. After using a red laser (635 nm) and a green laser (488 nm) to detect the IC16 CF633 and Nab228 CF488 antibodies, respectively, a blue laser (405 nm) was used, which does not target any specific antibody. By comparing the colocalized signals with the signals resulting from the blue laser, artificial autofluorescence signals can be detected and removed. This was performed by subtracting the number of autofluorescence pixels above the calculated cutoff from the number of colocalized pixels (corrected sFIDA readouts).

### Statistics
Statistical analyses were performed using Excel 2020 (Microsoft corporation, Redmond, USA), Matlab 2019b (The MathWorks, Natick, USA), OriginPro 2020 (OriginLab Corporation, Northhampton, USA) and python 3.9.7 (Python software foundation, Wilmington, USA; packages: pandas 1.3.4, scipy 1.7.1, seaborn 0.11.2).

All samples mentioned in the respective chapters were used for the analyses without further exclusion. As described above, each value is made up of at least 4 replicates.

**Intra-assay precision**. The PixelCount, coefficient of variation (CV %) and the sFIDA readout were calculated based on the mean value and standard deviation of the four replicates.

**Calibration**. Linear regression was performed for dilution experiments and to determine the concentration of aggregates in plasma samples. To this end, PixelCounts of the silica nanoparticles and the aggregate standards were weighted with 1/readout.

Corrected sFIDA readouts were used for calibration of DELCODE plasma samples: Based on these adjusted PixelCounts, the calibration of the samples was carried out using the SiNaPs dilution series between 0 and 125 fM for each of the six plates. After regression, the y-intercept of the regression models was subtracted from the respective calibrated values.

The limit of detection (LOD) and die lower limit of quantification (LLOQ) is described in Eqs. (1) and (2) and subsequently converted to a molar concentration using linear regression.

$$LOD[pixel] = PixelCount(blank\ control) + 1\sigma \tag{1}$$

$$LLOQ[pixel] = PixelCount(blank\ control) + 10\sigma \tag{2}$$

Analysis of the assay controls is mainly based on colocalization (sFIDA readout).

**Inter-assay precision.** Inter-assay precision was calculated among six individual experiments. Inter-assay variation for SiNaPs was calculated based on the mean coefficient of variation (CV %) for each concentration and thereafter by calculating the mean among the whole calibration curve. For inter-assay variation of aggregates and samples, we first determined the molar concentrations and afterwards proceeded as described above for inter-assay variation of SiNaPs. The potential significance of experimental differences was further examined by performing ANOVA of repeated measurements for SiNaPs, aggregates and samples with a 5% level of significance.

**Dilution linearity.** Two-fold dilutions were performed to analyse the influence of dilution on the calibration standard and a simulated plasma sample (aggregates spiked in plasma). sFIDA readouts were calibrated to molar concentrations and corrected by the dilution factor.

**Tube transfer.** For analysis of the effect of repeated tube transfers on sFIDA readout, 100 μl of plasma were transferred to a new tube, incubated for 5 min at RT and transferred once again (repeated according to the number of tube transfers).

**Spike and recovery.** A blank plasma sample (reference) and three plasma samples of Control, MCI or AD patients were analysed directly (unspiked) and after spiking with a low (31.3 fM), medium (250 fM) and high (2 pM) concentration of $A\beta_{1-15}$ SiNaPs. Recovery was calculated based on Eq. 3:

$$Recovery[\%] = \frac{sFIDA\ Readout_{spiked\ sample} - sFIDA\ Readout_{non-spiked\ sample}}{sFIDA\ Readout_{spiked\ BC} - sFIDA\ Readout_{non-spiked\ BC}}$$

(3)

**Pre-analytical and selectivity studies.** The effect of freeze-thaw cycles was evaluated using aliquots of six plasma samples, which were thawed repeatedly for 2 h at RT and frozen again[36]. The remaining signal (recovery) and signal reduction for sFIDA readouts of tube transfer, capture control (CC), autofluorescence control (AF), other probes (OP) and immunodepletion (IP) was calculated according to Eq. 4 and Eq. 5 directly, whereas PixelCounts were initially normalized with the blank control.

$$Recovery\ and\ remaining\ signal[\%] = \frac{sFIDA\ readout_{assay\ control}}{sFIDA\ readout_{reference}} * 100\%$$

(4)

$$signal\ reduction[\%] = 100\% - remaining\ signal[\%]$$

(5)

**Capture control.** For the capture control, no capture antibody was added in the first incubation step to analyze unspecific binding of the analyte to the assay surface. All other steps were performed as described in Assay setup. For comparison of capture control in plasma and buffer, a cutoff of 0.01% was chosen to reduce the influence of background noise from the different matrices. Capture control refers to the signal of the analyte compared to the assay setup with capture antibody according to Eq. (4).

**DELCODE plasma samples.** After calibration, samples below the LOD were set to zero. Calibrated concentrations were first tested for normal distribution. For non-normally distributed data, non-parametric tests like Mann-Whitney-U test and Spearman correlation tests were performed.

**Bootstrapping.** For testing the significance of the Spearman correlation, bootstrapping was applied to all samples with Aβ oligomer concentrations above LOD. This was achieved by performing 5000 replications of the bootstrapping with replacement and calculating the mean Spearman r value of the results. After normalization of the standard deviation, bootstrapping p-values were calculated using a normal distribution.

### Plasma samples

**Plasma of validation cohort.** Samples of Control, MCI and AD patients were kindly provided by the working group of Oliver Peters at Charité Berlin from patients. Plasma samples were centrifuged, aliquoted to 500 μl and stored at −80 °C. Samples did not undergo a freeze-thaw cycle prior to analysis. Written, informed consent was obtained from all participants. We obtained ethical approval from the Charité Berlin Institutional Review Board, approval number EA2/118/15.

**Plasma of DELCODE cohort.** Plasma samples were collected as part of the multicentre DZNE-Longitudinal Cognitive Impairment and Dementia Study (DELCODE) at ten clinical centres in Germany, according to a standard operating procedure. After processing, plasma samples were stored at −80 °C[37]. The project was approved by the ethical committee of the Charité Berlin (EA1/074/21 and EA4/066/17). Written, informed consent was obtained from all participants[37].

We received plasma samples from 429 patients, including healthy Controls (n = 44), SCD (n = 148), MCI (n = 92), AD patients (n = 52) and first-degree relatives of AD patients (n = 30). Sixty-two samples were excluded from analysis because of contamination in one experiment, one sample was excluded because of missing data and seven samples were excluded because of haemolysis. Besides testing cognitive function using different neuropsychological tests (i.e., mini mental state examination (MMSE), Alzheimer's disease assessment scale (ADAS), clinical dementia rating), CSF and plasma biomarkers like $A\beta_{1-40}$, $A\beta_{1-42}$, tau phosphorylated at threonine 181 (pTau) and total tau (tTau) were determined. Information on patient selection, sampling of blood and CSF, neuropathological tests and biomarker quantification are described in Jessen et al.[37]. Subdivision of patient groups by amyloid pathology is based on the $A\beta_{1-42}/A\beta_{1-40}$ ratio with previously established limits[38].

### Reporting summary
Further information on research design is available in the Nature Portfolio Reporting Summary linked to this article.

### Results
Initially, we validated the sFIDA assay using our $A\beta_{1-15}$-coated silica nanoparticle (SiNaPs) standard[28]. Additionally, Aβ aggregates were used to simulate a positive plasma sample, referred to as internal quality control (IQC). To this end, control plasma samples were spiked with different concentrations of Aβ oligomers. The synthesis and characterization of these aggregates is described in Pils et al. including a setup image in the supplement[32].

Additionally, we confirmed the sensitivity and selectivity of the assay for Aβ oligomers in a validation cohort comprising 20 plasma samples of control subjects (Control), mild cognitive impairment (MCI) and AD patients. Exemplary raw data images for the red and green fluorescence channels and colocalization are shown in Fig. 1b. We then applied the assay to a larger set of plasma samples of the DELCODE cohort, which comprises a control group, subjective cognitive decline (SCD), MCI, AD patients and first-degree relatives of AD patients (Table 1).

### sFIDA features high sensitivity and precision
**Analytical sensitivity.** Quantification of Aβ oligomers in plasma requires extreme sensitivity. Therefore, we initially investigated analytical sensitivity of sFIDA using $A\beta_{1-15}$-coated SiNaPs spiked in plasma and calculated a limit of detection (LOD) of 1.8 fM for the colocalization (Fig. 1c). Dilution linearity of SiNaPs was demonstrated between 2 fM to 8 pM with a mean dilution linearity of 100.6% and coefficient of

**Table 1 | Demographic information and biomarker concentrations of participants from the DELCODE cohort (mean ± standard deviation)**

|  | Controls | Relatives | SCD | MCI | AD |
|---|---|---|---|---|---|
| **Patient information** |  |  |  |  |  |
| number | 44 | 30 | 146 | 88 | 51 |
| % female | 45% | 63% | 42% | 45% | 67% |
| Age [years] | 68.7 ± 5.2 | 65.7 ± 5.0 | 70.8 ± 6.0 | 71.4 ± 5.0 | 75.9 ± 5.7 |
| Education [years] | 14.5 ± 2.5 | 14.4 ± 2.7 | 15.2 ± 2.8 | 13.6 ± 2.8 | 13.1 ± 3.1 |
| MMSE | 29.6 ± 0.6 | 29.2 ± 1.1 | 29.2 ± 1.1 | 27.5 ± 2.0 | 23.2 ± 3.3 |
| *ApoE* ε4 positive | 8 (18.2%) | 11 (36.7%) | 46 (31.5%) | 40 (46.0%)[a] | 31 (60.8%) |
| ε2/4 | 0 | 1 (3.3%) | 4 (2.7%) | 3 (3.4%)[a] | 2 (3.9%) |
| ε3/4 | 8 (18.2%) | 10 (33.3%) | 39 (26.7%) | 32 (36.8%)[a] | 21 (41.2%) |
| ε 4/4 | 0 | 0 | 3 (2.1%) | 5 (5.7%)[a] | 8 (15.7%) |
| **CSF biomarkers** |  |  |  |  |  |
| $A\beta_{1-40}$ [pg/ml] | 9321.1 ± 2617.7 | 8675.0 ± 2412.6 | 8679.5 ± 2213.3 | 8158.6 ± 2378.5[b] | 8179.5 ± 2475.2[a] |
| $A\beta_{1-42}$ [pg/ml] | 875.2 ± 344.4 | 903.6 ± 353.1 | 808.6 ± 355.3 | 604.7 ± 309.7[b] | 426.6 ± 211.9[a] |
| $A\beta_{1-42}/A\beta_{1-40}$ | 0.094 ± 0.024 | 0.103 ± 0.024 | 0.092 ± 0.028 | 0.075 ± 0.030[b] | 0.053 ± 0.018[a] |
| tTau [pg/ml] | 413.3 ± 185.1 | 333.0 ± 135.9 | 379.1 ± 194.3 | 532.5 ± 287.4[b] | 744.2 ± 344.2[a] |
| pTau [pg/ml] | 54.9 ± 22.6 | 49.8 ± 20.3 | 55.9 ± 24.1 | 69.7 ± 43.3[b] | 89.7 ± 34.1[a] |
| **Plasma biomarkers** |  |  |  |  |  |
| $A\beta_{1-40}$ [pg/ml] | 76.9 ± 18.7 | 74.6 ± 19.0 | 84.3 ± 20.0 | 86.1 ± 20.8[b] | 94.5 ± 28.0[a] |
| $A\beta_{1-42}$ [pg/ml] | 8.8 ± 1.9 | 8.5 ± 1.7 | 9.1 ± 1.9 | 8.5 ± 2.1[b] | 9.1 ± 2.5[a] |
| $A\beta_{1-42}/A\beta_{1-40}$ | 0.117 ± 0.022 | 0.115 ± 0.013 | 0.110 ± 0.015 | 0.099 ± 0.014[b] | 0.094 ± 0.019[a] |

*SCD* subjective cognitive decline, *MCI* mild cognitive impairment, *AD* Alzheimer's Disease.
[a]data not available for one patient.
[b]data not available for two patients.

determination of 0.994. Moreover, an upper limit of quantification (ULOQ) for SiNaPs was determined to be 256 pM, showing a 5-log dynamic range of sFIDA (Supplementary Fig. 2). For recombinant Aβ aggregates as IQC, an LOD of 1.9 pg/ml (monomer unit concentration) and a mean dilution linearity of 98.2% was calculated (Fig. 1d). Colocalization enhances the analytical sensitivity by two-fold for SiNaPs and ten-fold for recombinant aggregates compared to the individual channels red and green (Supplementary Fig. 3). Thus, colocalization was used unless otherwise stated. In a small proof-of-concept study including 20 plasma samples of healthy Control, MCI and AD patients, we measured Aβ oligomer concentrations ranging from 0 to 500 fM, confirming sufficient sensitivity of sFIDA for in vivo Aβ oligomers (sFIDA readouts in Supplementary Fig. 4a). Moreover, these oligomers showed a similar size distribution and amount of colocalization compared to synthetic Aβ-SiNaPs and aggregates (Fig. 1b). Pre-analytical studies indicated that tube transfers and freeze-thaw-cycles should be avoided (Supplementary Fig. 5).

**Intra-assay precision.** Mean intra-assay variation of SiNaPs among all concentrations was 9.4% for the red fluorescence channel (IC16 CF633), 4.9% for the green fluorescence channel (Nab228 CF488) and 15.5% for colocalization based on four replicates (Fig. 1c). Recombinant aggregates showed a mean intra-assay variation of 19.1% for the colocalization (Fig. 1d).

**Inter-assay precision.** Repeated measurements of SiNaPs spiked in plasma yielded a mean inter-assay variation of 19.3% for all concentrations tested (Fig. 3a). Calibrated concentrations of BC, two IQC and seven plasma samples showed a mean inter-assay variation of 41.9%. Using repeated measures ANOVA, the individual experiments for measurement of SiNaPs, IQC and plasma samples did not differ significantly (p-value > 0.05).

## Aβ oligomer quantification is not influenced by endogenous substances

**Recovery and dilution linearity.** We spiked three concentrations of SiNaPs in plasma samples from three individual patients to investigate matrix effects and calculated a mean percent recovery of 92% (excluding one concentration/sample) (Supplementary Fig. 6a, b). Additionally, the calibrated concentrations of SiNaPs and aggregates spiked in plasma were not affected by dilution with buffer (Supplementary Fig. 6c, d). The observed negligible effects of the sample matrix in both experiments showed that quantification of Aβ oligomers in individual plasma samples is accurate.

**Monomers.** In plasma, concentrations of approximately 300 pg/ml $A\beta_{1-40}$ and 20 pg/ml $A\beta_{1-42}$ have been determined[39]. Interference from Aβ monomers on sFIDA measurements was investigated by spiking 452 pg/ml Aβ monomers in a blank plasma sample. As a positive control, the same concentration of aggregated $A\beta_{1-42}$ was used. Monomer samples yielded a signal equivalent to the non-spiked blank control, whereas $A\beta_{1-42}$ aggregates yielded a nearly 100x stronger signal (Fig. 3d), indicating negligible interference from monomeric Aβ at physiologically relevant concentrations in our assay.

**HAMA.** In sandwich ELISAs, heterophilic antibodies (HA) can crosslink capture and detection antibodies, causing false-positive signals[30]. By changing the capture antibody Nab228, which gave false positive signals at concentrations of 10 ng/ml or higher (Fig. 3e), to bapineuzumab (humanized equivalent to 3D6[40]), interference from the spiked anti-mouse antibody was reduced to <0.005% at the highest concentration tested. Although a false-positive signal was observed at 1000 ng/ml HA, such concentrations are unlikely to be present in human plasma[41].

**Cross-reaction with αSyn and Tau aggregates.** We next investigated whether other protein aggregates composed of Tau or αSyn cross-reacted

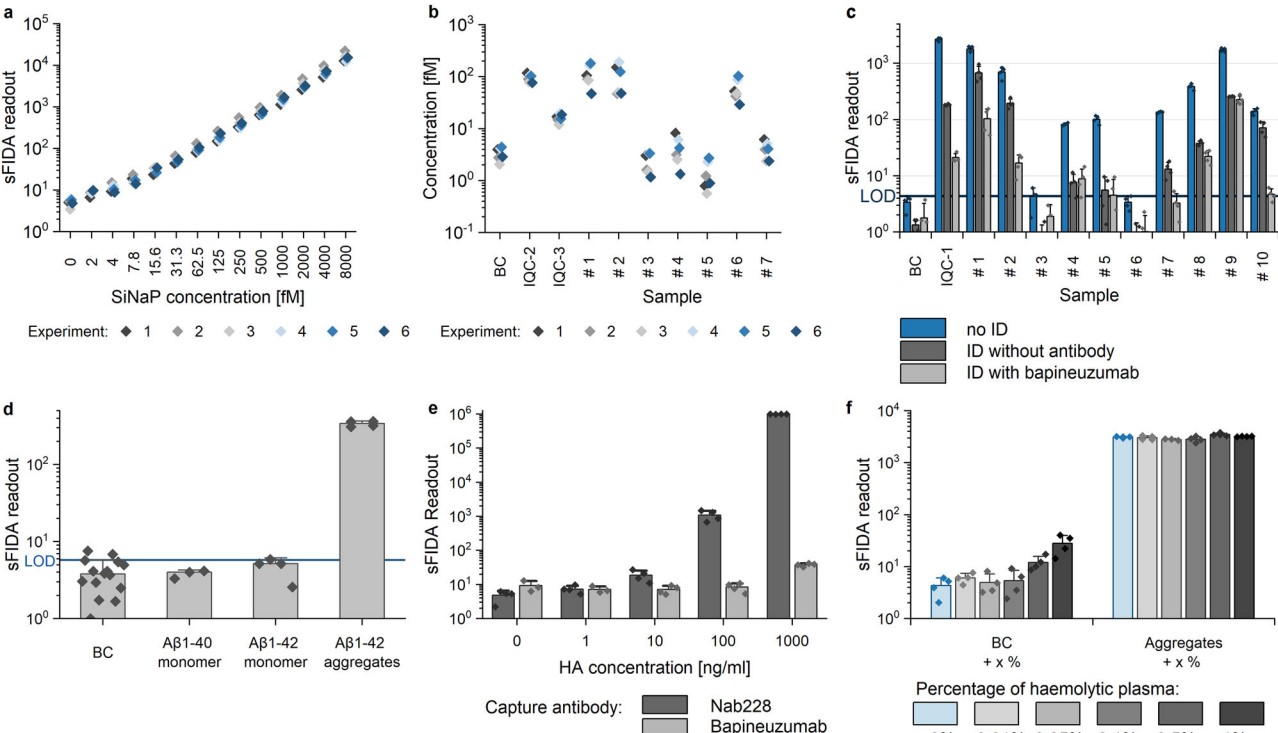

**Fig. 3 | Inter-assay variation and specificity controls for Aβ oligomer quantification in plasma. a** Repeated preparation of SiNaP calibration in six individual experiments showed an inter-assay variation of 19.3%. **b** Repeated measurements of seven samples of the validation cohort, a blank plasma (BC) and two internal quality controls (IQC, refers to aggregates at 141 pg/ml (IQC-2) and 17.6 pg/ml Aβ$_{1-42}$ monomers (IQC-3), respectively) were calibrated and mean inter-assay variation was calculated as 41.9%. **c** A blank control, Aβ$_{1-42}$ aggregates (IQC-1 with 18 ng/ml Aβ$_{1-42}$ monomer concentration) and 10 plasma samples of Control, MCI and AD patients were subjected to immunodepletion (ID). Unspecific ID (beads without antibody conjugation) resulted in a signal reduction to 6.9% for IQC-1 and to 20.2% for plasma samples for signals above the LOD (limit of detection). However, with specific immunodepletion using bapineuzumab, the signal of IQC-1 was eliminated

(signal <1% compared to the non-depleted sample) and that of the samples was reduced on average to 5.1%. **d** Blank plasma (BC) was spiked with 452 pg/ml of Aβ$_{1-40}$, Aβ$_{1-42}$ monomer and aggregates formed from 452 pg/ml Aβ$_{1-42}$ monomer. Samples were analysed by sFIDA. **e** Plasma was spiked with different concentrations of heterophilic antibody (HA) and analysed in two different assay setups, i.e., with monoclonal mouse antibody Nab228 as the capture antibody or with monoclonal humanized antibody bapineuzumab, respectively, to investigate heterophilic antibody interference. **f** Blank plasma and 18 ng/ml Aβ aggregates (concentration based on the monomer unit concentration) were spiked with haemolytic plasma and the effect on detection of aggregates was analysed. Standard deviations were calculated across the four replicates. Please, note the logarithmic scaling.

with our Aβ-specific detection system. The presence of αSyn or Tau aggregates gave no false-positive (BC spiked with αSyn or Tau aggregates) or false-negative signals (Aβ$_{1-42}$ aggregates spiked with αSyn or Tau aggregates), as all signals were ±20% from the non-spiked sample (Supplementary Fig. 4b).

**Haemolytic plasma.** During blood donation, erythrocytes may be damaged, leading to the release of haemoglobin through haemolysis[42]. As erythrocytes bind Aβ[43], their disruption may affect the sFIDA readout. This potential interference was examined by spiking a non-haemolytic sample with different concentrations of haemolytic plasma. Concentrations of 0.5% and 1% of haemolytic plasma produced a higher background signal, but did not affect aggregate detection (Fig. 2e). However, haemolytic plasma at a concentration of 0.5% gave a visibly reddish colour, indicating that these samples should be excluded from analysis.

## sFIDA readouts in plasma samples are solely attributed to Aβ oligomers

**Immunodepletion.** Immunodepletion was used to further demonstrate that the sFIDA-based Aβ oligomer signal does not originate from the plasma matrix. We removed Aβ species using magnetic beads coated with bapineuzumab, and used magnetic beads that were not coated with antibodies as a control. Non-specific immunodepletion reduced the signal for the recombinant aggregates (IQC-1) by 93.1% and on average by 79.8% for patient samples. In contrast, immunodepletion with

bapineuzumab reduced the signal for Aβ aggregates by 99.3% and that of the clinical samples by 94.9% for samples above LOD. Moreover, specific immunodepletion with bapineuzumab yielded a lower signal compared to unspecific immunodepletion for almost every sample tested with a mean signal reduction of 50.3% (Fig. 3c).

**Detection probe control.** The validation cohort was subjected to sFIDA in the absence of detection antibodies to exclude false-positive signals because of plasma sample autofluorescence. The autofluorescence signal was below the LOD for each sample tested with a mean signal reduction of >99% compared to the signal with detection antibodies (Supplementary Fig. 7a). Possible matrix interferences with IgG detection antibodies in general was investigated by probing plasma samples with an IgG isotype control (MOPC CF633) and an anti-αSyn antibody (211 CF488). The signals of the 20 plasma samples with these non-Aβ-specific probes were also reduced by >97% for the individual channels and colocalization (values of the colocalization are plotted in Supplementary Fig. 4a). Assay specificity was further increased by choosing an assay setup with two different anti-Aβ-probes and analysing only colocalized pixels.

**Capture control.** As non-specific adherence of Aβ oligomers to surfaces was reported previously[44], we investigated the unspecific binding of the analyte to the sFIDA assay surface by introducing a control where we skip the capture antibody (capture control). In the absence of a capture antibody, SiNaPs and aggregates spiked in plasma showed a signal of

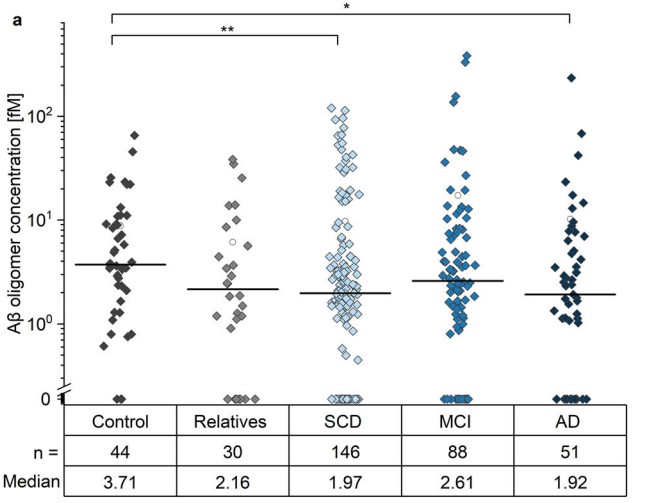

**Fig. 4 | Concentrations of Aβ oligomers in plasma samples. a** Aβ oligomer concentrations in plasma decreased significantly in SCD and AD patients compared to the Control group (*p* value SCD: 0.008; AD: 0.017). **b** After subdivision by amyloid pathology (A, based on CSF Aβ$_{1-42}$/Aβ$_{1-40}$ ratio[37]), SCD, MCI and AD patients positive for amyloid pathology (A +) showed significantly decreased Aβ oligomer concentrations in plasma compared to the amyloid negative (A-) Control group (*p* value for Control (A-) vs. amyloid positive SCD: 0.002; MCI: 0.041; AD: 0.031).

Please note the logarithmic scale. Samples that fell below LOD were set to zero. For reasons of clarity, the median Aβ oligomer concentrations are given for each analysis group below the logarithmically scaled figures (Median; values in fM). Number of samples in each group is referenced as n. SCD subjective cognitive decline, MCI mild cognitive impairment, AD Alzheimer's disease; open circle: mean; line: median, * *p*-value of Mann-Whitney-U test 0.01 - 0.05; ** *p*-value of Mann-Whitney-U test 0.001 - 0.01.

33.5% and 54.7%, respectively. However, the signals of SiNaPs and aggregates spiked in buffer were reduced by >99.9% in absence of a capture antibody (Supplementary Fig. 7b), indicating that surface binding is mediated by plasma matrix components. Similar to the calibration standards, five plasma samples tested showed non-specific binding of the analyte to the assay surface, indicated by a signal still at 77.2% (Supplementary Fig. 7c).

### Aβ oligomer concentrations in plasma samples are in the low femtomolar range

After analysing the sensitivity and selectivity of the analytical assay and investigating differences among the 20-sample validation cohort, we examined the disease-relevance and correlation with other AD-related biomarkers. Thus, we subjected 359 plasma samples from the DELCODE cohort to sFIDA analysis.

Like the validation cohort, Aβ oligomer concentrations determined in the samples from the DELCODE cohort spanned three orders of magnitude, ranging from 0.4 to 400 fM. Unexpectedly, AD (*p*-value: 0.017) and SCD (*p*-value: 0.008) subjects showed a significantly lower plasma Aβ oligomer concentration compared to control subjects using Mann-Whitney-U test, whereas samples of first-degree relatives and MCI patients did not show any significant differences (Fig. 4a). Interestingly, after subdividing groups according to their CSF amyloid pathology status (A + /A-, based on CSF Aβ$_{1-42}$/Aβ$_{1-40}$ ratio[37]), it became evident that oligomer concentrations are reduced in amyloid positive SCD, MCI and AD patients only (Fig. 4b, *p* value for Control (A-) vs. amyloid positive SCD: 0.002; MCI: 0.041; AD: 0.031).

### Correlations of Aβ oligomers in CSF and plasma depend on cognitive staging, amyloid pathology and *ApoE* status

We performed several correlation analyses to explore the pathophysiological basis for determining Aβ oligomer concentrations in plasma. We performed bootstrapping (re-sampling with replacement, see statistics) to enhance the reliability of our correlation analysis. Moreover, we grouped Control, relatives and SCD patients as well as MCI and AD patients for correlation analysis to enhance clarity and meaningfulness of our statistical analyses. No correlations of Aβ oligomers with Aβ monomers in plasma or with age or MMSE were observed (Supplementary Table 1). In contrast, Aβ

oligomers in plasma of MCI and AD patients showed a significant correlation with monomeric Aβ in CSF. We also analysed the correlation of Aβ oligomer concentrations in plasma versus CSF, which were recently measured by sFIDA[45]. Although oligomer concentrations in Control, relatives and SCD patients showed a direct correlation between CSF and plasma, oligomer concentrations in MCI and AD patients showed an inverse correlation (Fig. 5a). In AD, clearance of Aβ species from the brain is hypothesised to be impaired[2], which is probably dependent on the *ApoE* ε4 status or TREM-2 mutations affecting microglia activity[46,47]. Thus, we examined the dependency of Aβ oligomers in CSF and plasma on amyloid pathology or *ApoE* ε4 status. For both Control, relatives and SCD and MCI and AD patient groups, significant correlations were only observed for amyloid negative and *ApoE* ε4 negative patients, respectively. In contrast, when patients are *ApoE* ε4 positive or amyloid positive (A +), no significant correlations were found (Fig. 5b, c).

### Discussion

In the present study we adapted the sFIDA technology to quantify Aβ oligomers in human plasma samples. We demonstrated femtomolar sensitivity and low inter- and intra-assay variations for SiNaPs spiked in plasma. In contrast, plasma samples showed an increased inter-assay variation suggesting a yet unknown, possibly pre-analytical influence. However, taking into consideration the inherently high inter-assay variations at low concentrations[48], the 3-log difference between individual samples and the limited effect on the individual ranking of the samples, intra-assay variation was considered to be acceptable currently. Nevertheless, intra- and inter-assay variation may be improved in the future, by in-depth analyses of pre-analytical influences, and by applying full automation of the sFIDA assay to avoid human operator dependent variations, as has been partially applied previously[35].

Investigation of possible interfering factors, such as monomers and HAMA, and analysis of patient plasma samples from the validation cohort confirmed the sensitivity and selectivity of the sFIDA assay for quantification of Aβ oligomers. Although non-specific binding of Aβ oligomers to experimental surfaces did not influence the interpretation of the results, future efforts aim to reduce this issue to avoid signal loss in pre-analytical steps.

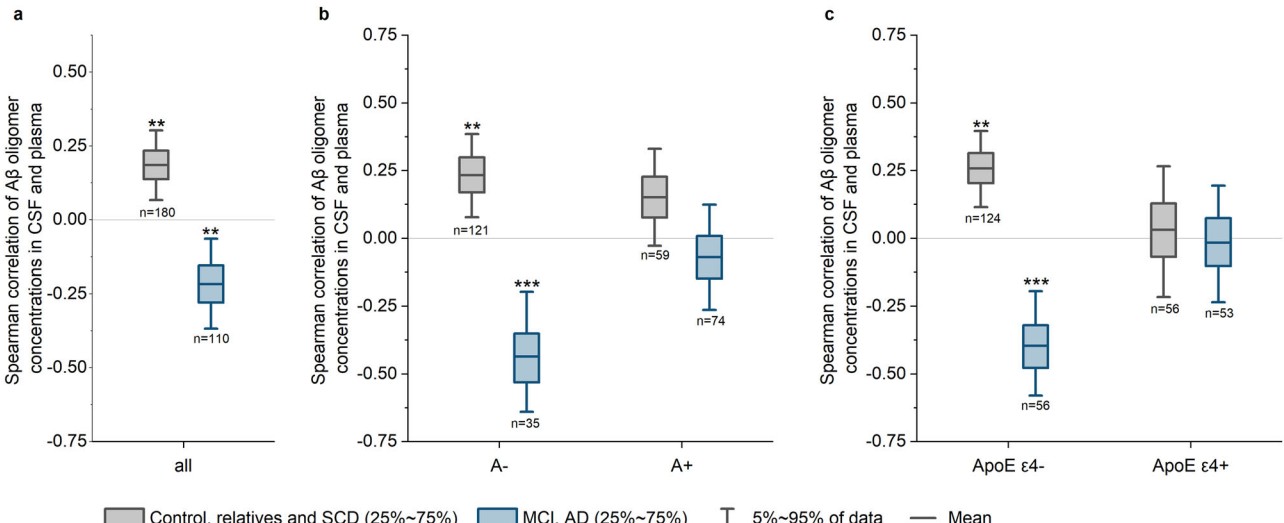

**Fig. 5 | Box plots for the bootstrap distribution of the Spearman coefficient of correlation r between Aβ oligomer levels in CSF and plasma. a** The combined group of Controls, relatives and SCD patients (grey) showed a weak, but significant direct correlation of Aβ oligomer levels in CSF and plasma (Spearman $r = 0.186$, $p$-value = 0.005), whereas MCI and AD patients (blue) showed an inverse correlation (Spearman $r = -0.217$, $p$-value = 0.009). **b** The groups were sub-divided by the presence of CSF amyloid pathology (A-/A +) based on the ratio of $A\beta_{1-42}/A\beta_{1-40}$. **c** The groups were sub-divided based on their *ApoE* ε4 status where carrying at least one *ApoE* ε4 allele defines positivity (*ApoE* ε4 +). Only for amyloid negative (A-) or *ApoE* ε4 negative patients, significant correlations between oligomers in CSF and plasma were observed with the Control, relatives and SCD patients showing a direct correlation (A-: Spearman $r = 0.233$, $p$-value = 0.007; ApoE ε4 negative: Spearman $r = 0.257$, $p$-value = 0.001) and MCI and AD patients an inverse correlation (A-: Spearman $r = -0.436$, $p$-value = <0.001; ApoE ε4 negative: Spearman $r = -0.396$, $p$-value = <0.001). Boxplots include 25-50% intervals with a line for the mean. Whiskers present the 5-95% intervals. $p$-value of Spearman $r$ distribution: * $p$-value 0.01 – 0.05, ** $p$-value 0.001 – 0.01, *** $p$-value < 0.001.

As the validation experiments showed the suitability of sFIDA to sensitively and specifically quantitate Aβ oligomers in plasma samples, we investigated Aβ oligomer concentrations in 359 plasma samples of the DELCODE cohort. Remarkably, we observed significantly reduced oligomer concentrations in SCD and AD patients compared to the Control group, which is in contrast to previous studies reporting increased Aβ oligomer concentrations in the plasma of AD patients[13,21,25] and a correlation of plasma Aβ oligomer concentration with SCD symptoms[49]. SCD is a heterogeneous condition with many potentially underlying causes – one of them is an early stage of AD[50]. In our cohort, 35.6% of SCD patients were amyloid positive and therefore fulfilling the NIA-AA research framework criteria for an underlying AD[51]. Although at a very early stage of AD, presumably the same mechanisms apply as with MCI and AD patients as discussed below. To interpret the differences to previous studies, it is important to point out, that most of these previous studies detected various oligomeric sub-species because of the use of structure-specific antibodies[17,24], detection of seeding-competent oligomers[21] or α-sheet content[25], whereas sFIDA quantifies the total amount of Aβ oligomers in plasma. It can be hypothesized that the Aβ oligomer subfractions examined by other studies might be subject to different formation and clearing mechanism compared to those described in the present study. Exploring whether differences in patient enrolment or pre-analytical aspects are responsible for these inconsistencies or alternative subpopulations of Aβ oligomers are measured by different assays is essential, and comparative studies using the same set of samples should be conducted. These varying outcomes across different assay setups emphasize the importance of such investigations.

This study aimed to quantify and better understand the potential origin of total Aβ oligomer concentrations in plasma samples. We observed that monomeric Aβ in plasma did not correlate with Aβ oligomers in plasma, whereas a correlation with CSF monomers and oligomers was observed. This observation indicates that Aβ oligomers, at least partially, originate from CSF (Fig. 6 clearance mechanism #4) or directly from the brain (Fig. 6 clearance mechanism #3). Therefore, it is possible that elevated oligomer concentrations in plasma may result from an increase in oligomer concentrations in CSF. Indeed, we observed a positive correlation of Aβ oligomers between CSF and plasma in Controls, relatives and SCD patients. However, this correlation was only evident for patients without amyloid pathology (A-, classification based on CSF $A\beta_{1-42}/A\beta_{1-40}$ ratio[37]) or without the genetic risk factor. We hypothesize that once amyloid positivity becomes evident (A +), Aβ oligomers are preferentially deposited in plaques (clearance mechanism #2), leading to a reduced clearance via other pathways (clearance mechanisms #1, #3, #4). This may explain the absence of a correlation between Aβ oligomers in the CSF and plasma, and the decrease in oligomer concentrations in the plasma of amyloid positive patients. Additionally, impaired clearance mechanisms for Aβ monomers across the BBB, BCSFB and perivascular drainage, and impaired degradation by microglia, have been reported for *ApoE* ε4 carriers (clearance mechanisms #1 and probably #3 and #4)[10,47]. Assuming similar pathological effects for oligomers, the most likely clearance mechanism in *ApoE* ε4 carriers is the deposition of Aβ oligomers into plaques (clearance mechanism #2), which limits the transport and results in a weaker correlation between CSF and plasma.

When interpreting correlations of Aβ oligomers between CSF and plasma in MCI and AD patients, it is important to consider previous sFIDA studies that have quantified Aβ oligomers in CSF. These studies have shown that Aβ oligomer concentrations in CSF are highest in the early stages of the disease and decrease as the disease progresses, particularly in *ApoE* ε4 carriers who have a higher Aβ oligomer burden during the early stages of the disease[45]. Decreasing concentrations of oligomers in CSF in advanced disease stages may arise from negative feedback mechanisms initiated by Aβ oligomers at synapses, resulting in reduced synaptic activity and consequently reduced production of Aβ monomers and replenishment of Aβ oligomers. Additionally, enhanced clearance through other pathways, such as deposition in plaques (clearance mechanism #2) or transport to the blood in a CSF-independent manner, may also contribute to reduced oligomer concentrations in CSF. We observed an inverse correlation of Aβ oligomer concentrations in CSF and plasma indicating an impaired clearance via CSF pathways (clearance mechanism #4) and an uncoupling of Aβ oligomer concentrations in blood and CSF. Moreover, this correlation was not observed for *ApoE* ε4 carriers, which supports the idea that *ApoE* ε4 plays a role in

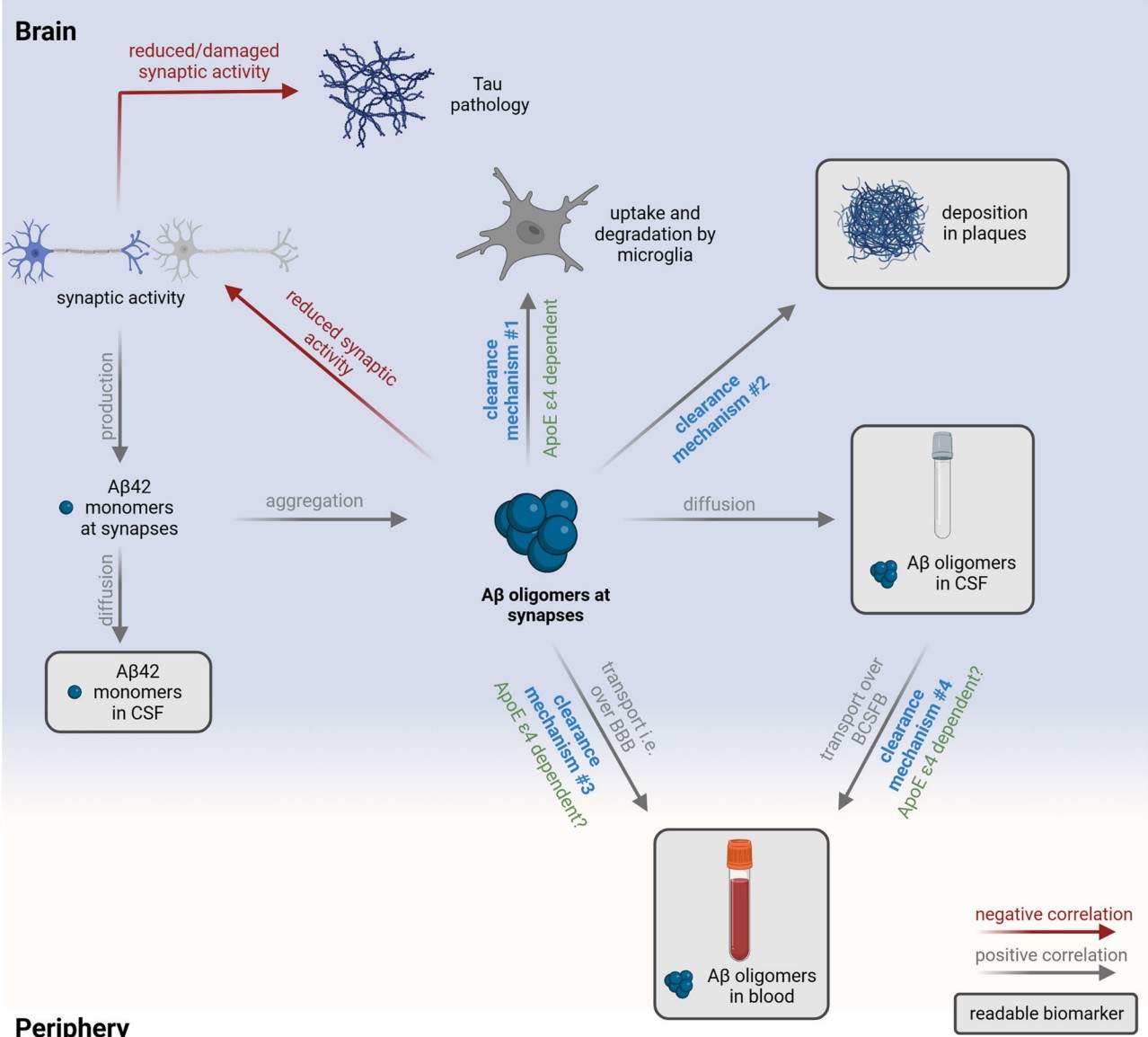

**Fig. 6 | Model of the clearance mechanisms for Aβ oligomers and the influence on the use of Aβ oligomers as biomarker.** Aβ monomer production at synapses is dependent on synaptic activity[55,56]. At a certain time point, aggregation of Aβ monomers leads to the formation of toxic Aβ oligomers that can be cleared by different mechanisms: Aβ oligomers can be degraded by microglia (clearance mechanism #1), diffuse into CSF or deposited into plaques (clearance mechanism #2). Moreover, Aβ oligomers may be transported to blood either directly across the BBB via glymphatic clearance or interstitial flow (clearance mechanism #3), or after diffusion into CSF and reaching blood via BCSFB (clearance mechanism #4). Formation of plaques in patients with amyloid pathology allows oligomers to be deposited (clearance mechanism #2), which may become the preferred fate of Aβ oligomers. This may lead to reduced clearance to blood and reduced Aβ oligomer concentrations in plasma. Additionally, transport of Aβ oligomers from the brain and CSF to plasma may be inefficient in *ApoE* ε4 carriers influencing correlation analysis. *Created with BioRender.com.*

transporting Aβ oligomers from the brain and thus an increase in oligomer concentrations in CSF for these carriers. However, these are only a few factors that influence Aβ oligomer clearance. Activation of microglia have also been associated with TREM-2 variants[46], which should be analyzed in future studies, as well as the weakly pronounced differences in oligomer concentrations and correlations, which has previously been observed in studies measuring plasma Aβ monomer concentrations[52]. Moreover, monomer concentrations in plasma were reported to depend on co-pathologies like hypertension, dyslipidemia, diabetes, liver function and chronic kidney disease[53]. These co-pathologies might be determinants of Aβ oligomer concentrations in plasma and should be considered in future studies.

Full interpretation of the results requires a better understanding of how Aβ oligomers are distributed and cleared from the brain and peripheral tissues during disease progression. Although we demonstrate here that the

sFIDA assay accurately measures total Aβ oligomer concentrations in plasma, the potential for using bloodborne Aβ oligomers as a diagnostic biomarker is limited due to the substantial overlap of individual readouts. Nevertheless, the statistically significant differences between the tested groups allow us to study the underlying pathophysiological role of Aβ oligomers. Owing to their central role in AD pathology, oligomers are a plausible therapeutic target to prohibit disease progression or even cure AD[54]. In pre-clinical and clinical development of anti-oligomer compounds, the quantification of plasma Aβ oligomers is a valuable tool to determine target engagement and to monitor therapeutic success at the molecular level.

## Data availability
The authors confirm that the data supporting the findings of this study are available within the article and its supplementary materials or can be made

available upon request. The source data for Figs. 1, 3, 4, 5, Supplementary Figs. 1, 2, 3, 4, 5 and 7 are included in the Supplementary Data file.

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

## Acknowledgements

We thank Dr. Carsten Korth (Institute of Neuropathology, Universitätsklinikum Düsseldorf, 40225 Düsseldorf, Germany) who kindly provided the IC16 antibody. We thank Dr. Volker Nischwitz (Central Institute of Engineering, Electronics and Analytics (ZEA-3), Forschungszentrum Jülich, 52428 Jülich, Germany) for ICP-MS measurements of silica nano-particles. We thank Dr. Andrew Dingley for proof-reding the manuscript. This work was supported by the Helmholtz Association [grant number HVF0079].

## Author contributions

L.B. developed the assay with support of M.P. L.B. performed the experiments and L.B. and F.R. analysed the data and carried out the statistics. V.K., A.C. and T.B. aid in experimental work and evaluation of the data. J.K. supported organization of the samples. L.B., O.B. and D.W. wrote the manuscript. O.B., D.W. and O.P. supervised the project. O.P., S.D.F., L.S.S., L.P., J.P., E.J.S., S.A., A.S., K.F., J.W., N.H., F.J., A.R., E.D., W.G., E.I.I., K.B., D.J., M.E., R.P., B.S.R., S.T., I.K., C.L., M.H.M., A.S., N.R., M.T.H., F.B., M.W., S.R., A.R., M.S. were responsible for overall design, implementation and collection of data for the DELCODE study at the respective study sites. All authors approved the final version of this manuscript.

## Funding

## Competing interests

D.W. and O.B. are co-founders and shareholders of attyloid GmbH. This affiliation had no influence of the interpretation of the data. All other authors declare no competing interests related to this work.

## Additional information

[1]Institute of Biological Information Processing (Structural Biochemistry: IBI-7), Forschungszentrum Jülich, 52428 Jülich, Germany. [2]attyloid GmbH, 40225 Düsseldorf, Germany. [3]German Center for Neurodegenerative Diseases (DZNE), 10117 Berlin, Germany. [4]Department of Psychiatry and Neuroscience, Campus Benjamin Franklin, Charité – Universitätsmedizin Berlin, corporate member of Freie Universität Berlin and Humboldt-Universität zu Berlin, 12203 Berlin, Germany. [5]Department of Psychiatry and Psychotherapy, Charité – Universitätsmedizin Berlin, corporate member of Freie Universität Berlin and Humboldt-Universität zu Berlin, 10117 Berlin, Germany. [6]School of Medicine, Technical University of Munich; Department of Psychiatry and Psychotherapy, 81675 Munich, Germany. [7]University of Edinburgh and UK DRI, Edinburgh, UK. [8]German Center for Neurodegenerative Diseases (DZNE), Bonn 53127 Bonn, Germany. [9]University of Bonn Medical Center, Dept. of Neurodegenerative Disease and Geriatric Psychiatry/Psychiatry, 53127 Bonn, Germany. [10]German Center for Neurodegenerative Diseases

(DZNE), 37075 Goettingen, Germany. [11]Department of Psychiatry and Psychotherapy, University Medical Center Goettingen, University of Goettingen, 37075 Goettingen, Germany. [12]Neurosciences and Signaling Group, Institute of Biomedicine (iBiMED), Department of Medical Sciences, University of Aveiro, 3810-193 Aveiro, Portugal. [13]Department of Psychiatry, University of Cologne, Medical Faculty, 50924 Cologne, Germany. [14]German Center for Neurodegenerative Diseases (DZNE), 39120 Magdeburg, Germany. [15]Institute of Cognitive Neurology and Dementia Research (IKND), Otto-von-Guericke University, 39106 Magdeburg, Germany. [16]Department for Psychiatry and Psychotherapy, University Clinic Magdeburg, 39120 Magdeburg, Germany. [17]German Center for Neurodegenerative Diseases (DZNE, Munich), 81377 Munich, Germany. [18]Institute for Stroke and Dementia Research (ISD), University Hospital, LMU Munich, 81377 Munich, Germany. [19]Department of Psychiatry and Psychotherapy, University Hospital, LMU Munich, 81377 Munich, Germany. [20]Munich Cluster for Systems Neurology (SyNergy) Munich, 81377 Munich, Germany. [21]Ageing Epidemiology Research Unit (AGE), School of Public Health, Imperial College London, London, UK. [22]Sheffield Institute for Translational Neuroscience (SITraN), University of Sheffield, Sheffield, UK. [23]Department of Neuroradiology, University Hospital LMU, 81377 Munich, Germany. [24]German Center for Neurodegenerative Diseases (DZNE), 18147 Rostock, Germany. [25]Department of Psychosomatic Medicine, Rostock University Medical Center, 18147 Rostock, Germany. [26]German Center for Neurodegenerative Diseases (DZNE), 72076 Tübingen, Germany. [27]Section for Dementia Research, Hertie Institute for Clinical Brain Research and Department of Psychiatry and Psychotherapy, University of Tübingen, 72076 Tübingen, Germany. [28]Department of Psychiatry and Psychotherapy, University of Tübingen, 72076 Tübingen, Germany. [29]University of Bonn Medical Center, Dept. of Neurology, 53217 Bonn, Germany. [30]Luxembourg Centre for Systems Biomedicine (LCSB), University of Luxembourg, L-4367 Belvaux, Luxembourg. [31]Excellence Cluster on Cellular Stress Responses in Aging-Associated Diseases (CECAD), University of Cologne, 50931 Cologne, Germany. [32]Division of Neurogenetics and Molecular Psychiatry, Department of Psychiatry and Psychotherapy, Faculty of Medicine and University Hospital Cologne, University of Cologne, 50931 Cologne, Germany. [33]Department of Psychiatry & Glenn Biggs Institute for Alzheimer's and Neurodegenerative Diseases, San Antonio, TX, USA. [34]Institute for Medical Biometry, University Hospital Bonn, 53127 Bonn, Germany. [35]Institut für Physikalische Biologie, Heinrich-Heine-Universität Düsseldorf, 40225 Düsseldorf, Germany.
✉e-mail: d.willbold@fz-juelich.de

