## [Peer Review file · Communications Medicine]

Blood-based quantification of A β oligomers indicates impaired clearance from brain in ApoE ϵ 4 positive subjects

Corresponding Author: Professor Dieter Willbold

Version 0:

Reviewer comments:

Reviewer #1

(Remarks to the Author)

In the manuscript entitled " Blood-based quantification of A β oligomers indicates impaired clearance from brain in ApoE ϵ 4 positive subjects" by Lara Blömeke et al., the authors addressed a need in the field of Alzheimer's Disease (AD) research, the development of a high-sensitivity and selectivity technology for quantifying A β oligomers in plasma. The surface-based fluorescence intensity distribution analysis (sFIDA) assay that they developed and validated demonstrates a remarkable sensitivity, with a detection limit of 1.8 fM. This achievement is noteworthy, considering the intricate nature of A β oligomer quantification in plasma. However, there are several concerns regarding the rationale, methods and interpretation of the results that are difficult to overlook. Please see below for comments.

Major

Introduction

While the introduction section outlines the role of A β oligomers as biomarkers and the development of an sFIDA assay, it falls short in clearly stating the study's specific research question or objective. It does not delve into the existing knowledge and controversies surrounding A β oligomers. An elaboration on the extensive diversity of A β oligomers (ADDL, spherical oligomer, globulomer, large oligomer, protofibrils...) and the multiple perspectives regarding their deleterious effects would enhance the foundation for readers less familiar with this field. Authors should also state more detailed knowledge about past literature measuring A β oligomers in plasma (or CSF), and mention what is the weak point of those published methods. In other word, author need to state what is the strength of the current sFIDA assay, and its relevance to clinical practice. Consequently, a compelling justification for the adoption of the sFIDA approach, backed by the perceived shortcomings of prior techniques (if any), should be presented.

Methods:

In consideration of readers who may be beginners to this specialized domain, the term "internal quality control" and the significance of "SiNaPs standards" should be explicated more comprehensively. Putting a set-up image (aside from current Fig1) would be helpful.

In the context of potential future clinical application, please provide a dedicated figure illustrating the detailed time course of the assay (required time), along with the capacity for simultaneous analysis of multiple samples.

Results

With regard to the "DELCODE" study, the authors identified elevated plasma A β oligomer concentrations in control group in comparison to the SCD group, while acknowledged as an unexpected outcome, still demands further elucidation.

Regarding the reference to "amyloid pathology" on lines 227-228, and 286-287. Are these same items? A more detailed description or explanation should be provided.

Although the authors assert that the sFIDA method quantifies the total amount of A β oligomers in plasma (Line 272), the rationale to focus on the total quantity of oligomers may hold greater significance in comparison to distinct oligomer subtypes warrants further explanations.

Aggregate Preparation and Characterization:

While the protocol for aggregate preparation of A β 42, α -synuclein, and Tau is meticulously detailed based on the past

literature, how the state of aggregation was validated or characterized would enhance the study's robustness. Validation data such as electron microscopy could contribute valuable insights into the morphology of aggregates. Also, why did the authors omit A β 40?

Minor

Abstract

Line 64 and 67; "A β " and "sFIDA", full spell is needed.

Introduction

Line 80 and 83; "A β " and "CSF", full spell is needed.

Discussion

Line 302-303; Authors might want to re-write as "unpublished observation".

Table 1.

"ApoE ϵ 4 positive"; if available, adding the number of ϵ 3/4, ϵ 4/4 cases will be more informative.

Reviewer #2

(Remarks to the Author)

The paper by Biomeke et al addresses an interesting issue as the measure of A β oligomers concentration in plasma associated with cognitive decline. The authors proposed a sFIDA assay that can determine a very low concentration (fmols) of oligomers in blood. The authors correctly analyzed the possible interference with the method by the plasma proteins and antibodies and besides the other point mentioned below, apparently the main problem with this method is the intra- (19%) and inter (41.9%) variability. In the discussion section the authors "...considered acceptable currently.." this variability and propose a method to improve the results but there is no explanation about why this method has not been applied.

The results in Fig.3 indicate a minimal but significant reduction of A β oligomer levels in AD, however although the subjects are numerous the effect is modest (the results in numbers together with the diagrams might help to understand the differences) and the comparison with relatives was not significant indicating a congenital difference rather than a consequence of the disease. As mentioned by the authors this result is in contrast with previous findings. All together these considerations make difficult to consider the results convincing and deserve of publication although I can appreciate the effort of possible explanation (described in scheme of Fig.5) in association with ApoE. At least the comparison with another cohort of AD subjects and with a cohort of PD subjects (in this case the plasma A β oligomer levels should be unchanged) are required to consider the results sufficiently robust.

Other points

In fig.1 the micropictures in b are not informative at all, the quality of the pictures must be improved to permit the identification of the fluorescent signal

All the paragraph dedicated to HAMAS can be summarized in the last two sentences

The paragraph "capture control" is incomprehensible to me, better explanations might help.

In the description of results, p.9 line 223, the correct sentence should be: "Unexpectedly, AD and SCD subjects showed a significantly reduction plasma A β oligomer concentrations compared to controls..." not the opposite

Version 1:

Reviewer comments:

Reviewer #1

(Remarks to the Author)

The authors responded to the comments properly, and no comments were made regarding the revised manuscript.

Reviewer #2

(Remarks to the Author)

The paper by Blömeke et al proposed a new method for the measure A β oligomers in plasma. As pointed out by the authors the results on the relationship between the levels A β oligomers and the cognitive status found in the cohort investigated with this method are different from previous association results (decrease vs increase), the explanation of this discrepancy (total oligomers vs specific oligomers) is not convincing. But the main problem of the paper is the modest differences found among the clinical groups and the up and down from SCD to MCI and AD or even control vs relative group (fig.3a) although some statistical significances are reported, the distribution of the values is substantially similar in each group. Furthermore in Fig 3b since both group are divided in A+ and A- the comparison should be done according to this stratification (SCD A+ vs control A+). The potential contribution of these determinations in diagnosis

appears modest. In terms of comprehension of oligomers production and distribution, the data in CSF would be very useful, they are mentioned as "not published yet" throughout the paper, my suggestion is to publish in a unique issue both results in CSF and plasma to give adequate information.

Reviewer #1 (Remarks to the Author):

In the manuscript entitled " Blood-based quantification of A β oligomers indicates impaired clearance from brain in ApoE ϵ 4 positive subjects" by Lara Blömeke et al., the authors addressed a need in the field of Alzheimer's Disease (AD) research, the development of a high-sensitivity and selectivity technology for quantifying A β oligomers in plasma. The surface-based fluorescence intensity distribution analysis (sFIDA) assay that they developed and validated demonstrates a remarkable sensitivity, with a detection limit of 1.8 fM. This achievement is noteworthy, considering the intricate nature of A β oligomer quantification in plasma. However, there are several concerns regarding the rationale, methods and interpretation of the results that are difficult to overlook. Please see below for comments.

Major

Introduction

Comment 1

While the introduction section outlines the role of A β oligomers as biomarkers and the development of an sFIDA assay, it falls short in clearly stating the study's specific research question or objective. It does not delve into the existing knowledge and controversies surrounding A β oligomers. An elaboration on the extensive diversity of A β oligomers (ADDL, spherical oligomer, globulomer, large oligomer, protofibrils...) and the multiple perspectives regarding their deleterious effects would enhance the foundation for readers less familiar with this field. Authors should also state more detailed knowledge about past literature measuring A β oligomers in plasma (or CSF), and mention what is the weak point of those published methods. In other word, author need to state what is the strength of the current sFIDA assay, and its relevance to clinical practice. Consequently, a compelling justification for the adoption of the sFIDA approach, backed by the perceived shortcomings of prior techniques (if any), should be presented.

We thank the reviewer to point out the diversity of oligomer species and giving us the opportunity to explain the advantages of the sFIDA technology. To address the points raised above, we elaborated the introduction in more detail (line 88 ff; Current wording in blue, new wording in green):

Although the exact clearance mechanisms of A β oligomers from the brain and CSF to plasma remain largely unknown, earlier studies have confirmed the presence of A β oligomer species in plasma samples ¹.

Disease progression may lead to a reduction in A β oligomers in plasma samples because of their deposition in amyloid plaques and impaired clearance from the brain into the blood stream ². For example, an inverse correlation between efficiency of glymphatic clearance and oligomer size has been described ³. Additionally, transport of A β across the BBB is affected in AD patients, especially in carriers of the AD risk gene allele apolipoprotein E (ApoE) ϵ 4 ^{3,4}. Quantification of A β oligomer concentrations in plasma samples, especially in early disease stages, and in-depth analysis of dependencies between A β oligomers and different biomarkers will improve our understanding of the role of amyloid pathology in AD.

Previous studies have reported higher oligomer concentrations in AD patients ^{2,5-7}. All of the methods applied in these studies detect specific subtypes of A β oligomers, depending on the respective antibody used. For A β oligomers derived from the brain, a broad range of species was described, ranging from small molecular weight oligomers like dimers and trimers via 56mers and spherical oligomers like A β derived diffusible ligands (ADDLs) to high-molecular weight oligomers and protofibrils ^{8,9}. For these species, differences have been claimed for their neurotoxicity and pathologic mechanisms, like impairment of mitochondrial dysfunction, Ca²⁺ homeostasis dysregulation and induction of tau pathology ⁸.

The most widely described method for detection of A β oligomers in plasma is the multimer detection system (MDS) which uses A β 1-42 to amplify the signal and is therefore a tool to measure the oligomerization tendency of on-pathway oligomers. Using this method, AD patients showed significantly increased MDS signal compared to the control group (An 2017, Meng 2019, Wang 2017) and the correlation with cognitive decline using neuropsychological tests like MMSE and CERAD (¹⁰⁻¹²). Other methods used oligomer specific antibodies to quantify those oligomer species that are recognized by the respective

specific antibody (^{1,7,13}). A third method claimed to quantify the alpha-sheet content of oligomers in plasma using specifically designed alpha-sheet peptides ¹⁴.

In contrast to these methods, the surface-based fluorescence intensity distribution analysis (sFIDA) technology aims to quantify all oligomer species, irrespective of their conformation, morphology and size, all of them potentially relevant for disease development and progression. sFIDA is a versatile platform for quantification of protein aggregates in biofluids that features single particle sensitivity due to a microscopy-based readout and selectivity for aggregated A β because of the use of antibodies with overlapping or even identical linear epitopes at the N-terminus of A β (principle of sFIDA in Fig. 1a). Quantifying the total amount of oligomeric species is crucial for quantitation of target engagement in the development of anti-oligomeric drugs. New therapies aim to eliminate A β oligomers. Using a diagnostic tool that captures all oligomer species may show the effect of this anti-oligomeric drug irrespective of the exact mechanism of action and the target oligomer species.

As calibration standard for oligomer-based assay, we previously established protein conjugated silica nanoparticles (SiNaPs) ¹⁵⁻¹⁷. Additionally, we demonstrated that sFIDA sensitively and specifically detects alpha synuclein (α Syn), Tau and A β oligomers in CSF samples ^{18,19}. Nonetheless, the reliable quantification of A β oligomers in plasma samples poses an even greater challenge, as plasma typically contains a 200-fold higher total protein concentration than CSF ²⁰. This high protein background can lead to false negative readouts because of epitope masking, or false positive readouts because of interferences with human anti-mouse-antibodies ²¹. Moreover, A β oligomer concentrations are expected to be in the low femto- to even attomolar range ²², thus requiring an extremely sensitive method for detection.

In this report, an sFIDA assay for quantification of A β oligomers in plasma samples was developed and validated as a basic research project. We intended to quantitate the total A β oligomer levels in plasma samples to investigate the development of their concentrations during disease progression and their dependency from the ApoE4 status of the donors.

Methods:

Comment 2

In consideration of readers who may be beginners to this specialized domain, the term "internal quality control" and the significance of "SiNaPs standards" should be explicated more comprehensively. Putting a set-up image (aside from current Fig1) would be helpful.

We thank the reviewer for this proposal. We created a new supplementary Figure, which describes the synthesis and principle of the SiNaP calibration standard (Supplementary fig. 6, reference in the manuscript in line 388 (methods)). For the aggregates, a setup image can be viewed in the supplement of the manuscript of Pils et al.²³. We added a statement in the beginning of the results section (line 141) to refer to this manuscript and its supplement and added the image here for your information.

SiNaPs:

Supplementary Fig. 5 Synthesis of protein conjugated SiNaPs. The process of protein conjugation of SiNaPs consists of three main steps. First, the silica core is functionalized with APTES. As a second step, MIHA is added as a crosslinker between the SiNaPs core and the protein. The use of maleimide as functional group in combination with the reaction conditions allows a directed coupling with the thiol group at the modified C-terminus of the protein. The resulting protein conjugated SiNaPs imitate a protein aggregate with multiple binding sites for the antibody but showing a unique size distribution, high stability, and defined epitope number ¹⁶.

Aggregates:

Additionally, A β aggregates were used to simulate a positive plasma sample, referred to as “internal quality control” (IQC). To this end, control plasma samples were spiked with different concentrations of A β oligomers. The synthesis and characterization of these aggregates is described in Pils et al. including a setup image in the supplement ²³.

Setup image of synthesis of A β aggregates (from Pils et al. ²³).

Comment 3:

In the context of potential future clinical application, please provide a dedicated figure illustrating the detailed time course of the assay (required time), along with the capacity for simultaneous analysis of multiple samples.

We thank the reviewer for this comment. We added a figure (Figure 6) to the Methods section (line 438) providing an overview about the assay steps, the time course of the individual steps as well as total time and the capacity for simultaneous analysis of samples.

Fig. 6 sFIDA workflow. The use of 384 well plates allow a close-meshed concentration series and the determination of 79 patient samples on one plate in 4-fold replicate determination. The individual steps consist of an over night (ON) incubation of the capture antibody at 4°C, a 1.5 h blocking step at room temperature (RT) followed by 2 h incubation of the plasma samples and 1 h incubation of detection antibodies. The final measurement is conducted by an automated fluorescence microscope. *Created with BioRender.com*

Results

Comment 4

With regard to the "DELCODE" study, the authors identified elevated plasma A β oligomer concentrations in control group in comparison to the SCD group, while acknowledged as an unexpected outcome, still demands further elucidation.

We extended the explanation in line 293 ff.:

Remarkably, we observed significantly reduced oligomer concentrations in SCD and AD patients compared to the control group, which is in contrast to previous studies reporting increased A β oligomer concentrations in the plasma of AD patients^{2,14,24} and a correlation of plasma A β oligomer concentration with SCD symptoms²⁵. SCD is a heterogeneous condition with many potentially underlying causes – one of them is an early stage of AD²⁶. In our cohort, 35.6% of SCD patients were amyloid positive and therefore fulfilling the NIA-AA research framework criteria for an underlying AD²⁷. Although at a very early stage of AD, presumably the same mechanisms apply as with MCI and AD patients as discussed below. To interpret the differences to previous studies, it is important to point out, that most of these previous studies detected various oligomeric sub-species because of the use of structure-specific antibodies [...]

Comment 5

Regarding the reference to “amyloid pathology” on lines 227-228, and 286-287. Are these same items? A more detailed description or explanation should be provided.

We thank the reviewer for this comment. For classifying the patients in A- and A+, we used the CSF $A\beta_{1-42}/A\beta_{1-40}$ ratio for all of the analysis. To clarify this in the manuscript, we added the statement “based on CSF $A\beta_{1-42}/A\beta_{1-40}$ ratio” in both lines as well as in figure legend 3b.

- Former line 228 (now line 250) (A+/A-, based on CSF $A\beta_{1-42}/A\beta_{1-40}$ ratio ²⁸)
- Former line 287 (now line 316): (classification based on CSF $A\beta_{1-42}/A\beta_{1-40}$ ratio ²⁸)
- Fig. 3 b: After subdivision by amyloid pathology (A, based on CSF $A\beta_{1-42}/A\beta_{1-40}$ ratio ²⁸)

Comment 6

Although the authors assert that the sFIDA method quantifies the total amount of A β oligomers in plasma (Line 272), the rationale to focus on the total quantity of oligomers may hold greater significance in comparison to distinct oligomer subtypes warrants further explanations.

We thank the reviewer for this comment. In the revised introduction based on comment 1, we now added a rationale why the total amount of oligomers is interesting to consider. Please, refer to comment 1 for further explanations.

Comment 7

Aggregate Preparation and Characterization:

While the protocol for aggregate preparation of A β 42, α -synuclein, and Tau is meticulously detailed based on the past literature, how the state of aggregation was validated or characterized would enhance the study's robustness. Validation data such as electron microscopy could contribute valuable insights into the morphology of aggregates.

We thank the reviewer for this comment. The aggregation protocols used were based on the methods described in previous studies including the characterization of the aggregates. These experiments were not repeated for this study. However, to give the readers the opportunity to inform themselves, we added information about the methods that were used to characterize the aggregates.

- A β aggregates (line 411 ff): [...] A β aggregates have previously been characterized in Pils et al. ²³ using Thioflavin T assay (THT) and atomic force microscopy (AFM). Aggregates showed a monodisperse size distribution with a mean diameter of 2.7 nm.
- α -Syn aggregates (line 419 ff): [...] Preparation of α -Syn aggregates was based on Lohmann et al. 2019 ²⁹. For characterization, AFM measurements were used ²⁹.
- Tau aggregates (line 426 ff): [...] Tau aggregates were characterized previously including AFM measurement and THT ³⁰.

Also, why did the authors omit A β 40?

We have added the following sentences to the Methods part on A β aggregate preparation (page 17, line 313):

We focused on A β ₁₋₄₂ to prepare artificial aggregates, because in our hands their preparation is more robust and reproducible, and, because the capture and detection antibodies used here do not discriminate between A β ₁₋₄₀ and A β ₁₋₄₂.

Minor

Abstract

Line 64 and 67; “A β ” and “sFIDA”, full spell is needed.

Done

Introduction

Line 80 and 83; “A β ” and “CSF”, full spell is needed.

Done

Discussion

Line 302-303; Authors might want to re-write as “unpublished observation”.

We changed the wording in line 262 and 333 to “yet unpublished observation”.

Table 1.

“ApoE ϵ 4 positive”; if available, adding the number of ϵ 3/4, ϵ 4/4 cases will be more informative.

	Controls	Relatives	SCD	MCI	AD
Patient information					
number	44	30	146	88	51
% female	45%	63%	42%	45%	67%
Age [years]	68.7 \pm 5.2	65.7 \pm 5.0	70.8 \pm 6.0	71.4 \pm 5.0	75.9 \pm 5.7
Education [years]	14.5 \pm 2.5	14.4 \pm 2.7	15.2 \pm 2.8	13.6 \pm 2.8	13.1 \pm 3.1
MMSE	29.6 \pm 0.6	29.2 \pm 1.1	29.2 \pm 1.1	27.5 \pm 2.0	23.2 \pm 3.3
ApoE ϵ 4 positive	8 (18.2%)	11 (36.7%)	46 (31.5%)	40* (46.0%)	31 (60.8%)
ϵ 2/4	0	1 (3.3%)	4 (2.7%)	3* (3.4%)	2 (3.9%)
ϵ 3/4	8 (18.2%)	10 (33.3%)	39 (26.7%)	32* (36.8%)	21 (41.2%)
ϵ 4/4	0	0	3 (2.1%)	5* (5.7%)	8 (15.7%)
CSF biomarkers					
A β ₁₋₄₀ [pg/ml]	9321.1 \pm 2617.7	8675.0 \pm 2412.6	8679.5 \pm 2213.3	8158.6 \pm 2378.5	8179.5 \pm 2475.2
A β ₁₋₄₂ [pg/ml]	875.2 \pm 344.4	903.6 \pm 353.1	808.6 \pm 355.3	604.7 \pm 309.7	426.6 \pm 211.9
A β ₁₋₄₂ /A β ₁₋₄₀	0.094 \pm 0.024	0.103 \pm 0.024	0.092 \pm 0.028	0.075 \pm 0.030	0.053 \pm 0.018
tTau [pg/ml]	413.3 \pm 185.1	333.0 \pm 135.9	379.1 \pm 194.3	532.5 \pm 287.4	744.2 \pm 344.2
pTau [pg/ml]	54.9 \pm 22.6	49.8 \pm 20.3	55.9 \pm 24.1	69.7 \pm 43.3	89.7 \pm 34.1

Plasma biomarkers

A β ₁₋₄₀ [pg/ml]	76.9 ± 18.7	74.6 ± 19.0	84.3 ± 20.0	86.1 ± 20.8	94.5 ± 28.0
A β ₁₋₄₂ [pg/ml]	8.8 ± 1.9	8.5 ± 1.7	9.1 ± 1.9	8.5 ± 2.1	9.1 ± 2.5
A β ₁₋₄₂ /A β ₁₋₄₀	0.117 ± 0.022	0.115 ± 0.013	0.110 ± 0.015	0.099 ± 0.014	0.094 ± 0.019

*data not available for one patient

Reviewer #2 (Remarks to the Author):

Comment 1

The paper by Blomeke et al addresses an interesting issue as the measure of A β oligomers concentration in plasma associated with cognitive decline. The authors proposed a sFIDA assay that can determine a very low concentration (fmoles) of oligomers in blood. The authors correctly analyzed the possible interference with the method by the plasma proteins and antibodies and besides the other point mentioned below, apparently the main problem with this method is the intra- (19%) and inter (41.9%) variability. In the discussion section the authors "...considered acceptable currently." this variability and propose a method to improve the results but there is no explanation about why this method has not been applied.

We thank the reviewer for revealing this misunderstanding. In fact, the potentially improved method was not available. To avoid this misunderstanding, we have modified the respective sentence (page 11, line 280):

In contrast, plasma samples showed an increased inter-assay variation suggesting a yet unknown, possibly pre-analytical influence. However, taking into consideration the inherently high inter-assay variations at low concentrations³¹, the 3-log difference between individual samples and the limited effect on the individual ranking of the samples, intra-assay variation was considered to be acceptable currently. Nevertheless, intra- and inter-assay variation may be improved in the future, by in-depth analyses of pre-analytical influences, and by applying full automation of the sFIDA assay to avoid human operator dependent variations, as has been partially applied previously³².

Comment 2

A) The results in Fig.3 indicate a minimal but significant reduction of A β oligomer levels in AD, however although the subjects are numerous the effect is modest (the results in numbers together with the diagrams might help to understand the differences)

Fig. 1 Concentrations of A β oligomers in plasma samples. **a** A β oligomer concentrations in plasma decreased significantly in SCD and AD patients compared to the cognitively normal control group. **b** After subdivision by amyloid pathology (A), SCD, MCI and AD patients positive for amyloid pathology (A+) showed significantly decreased A β oligomer concentrations in plasma compared to the amyloid negative (A-) control group. Please note the logarithmic scale. Samples that fell below LOD were set to zero. For reasons of clarity, the median A β oligomer concentrations are given for each analysis group below the logarithmically scaled figures (Median; values in fM). SCD: subjective cognitive decline; MCI: mild cognitive impairment; AD: Alzheimer's disease; open circle: mean; line: median, * p value of Mann-Whitney-U test 0.01 - 0.05; ** p value of Mann-Whitney-U test 0.001 - 0.01.

B) and the comparison with relatives was not significant indicating a congenital difference rather than a consequence of the disease.

We appreciate the reviewer's suggestion of a congenital difference rather than a consequence of the disease. Similarities between diseased subjects and relatives, however, may indicate similar congenital backgrounds, but also similarities in living environment and/or life style. Thus, we refrained from any further speculations on any similarities or differences regarding the relatives group. This certainly would need further investigations.

- C) As mentioned by the authors this result is in contrast with previous findings. All together these considerations make difficult to consider the results convincing and deserve of publication although I can appreciate the effort of possible explanation (described in scheme of Fig.5) in association with ApoE.

We very much appreciate this comment. Our results and controls confirm the suitability of sFIDA in measuring A β oligomer plasma levels. However, we are aware of the discrepancies with earlier published research, which might stem from variations in the methodologies used. Unlike other techniques that identify only a subset of A β oligomer species, sFIDA captures all A β oligomers regardless of their size and morphology. It can be hypothesized that the A β oligomer subfractions examined by other studies might be subject to different formation and clearing mechanism compared to those described in the present study. We extended our discussion with the last sentence (line 301): “It can be hypothesized that the A β oligomer subfractions examined by other studies might be subject to different formation and clearing mechanism compared to those described in the present study.”

- D) At least the comparison with another cohort of AD subjects and with a cohort of PD subjects (in this case the plasma A β oligomer levels should be unchanged) are required to consider the results sufficiently robust.

We thank the reviewer for the idea of measuring different disease cohorts to confirm the results of this study. However, as often in neurodegenerative diseases, there are so many co-pathologies, e.g. common amyloid pathology in PD patients³³ that we do not expect a clear distinction as if PD patients would have no Abeta oligomers in blood. This, however, is certainly something we will investigate in the future.

Other points

In fig.1 the micropictures in b are not informative at all, the quality of the pictures must be improved to permit the identification of the fluorescent signal.

We thank the reviewer for this note. We have enlarged a section of Fig. 1 to improve the identification of the fluorescence signal.

Fig. 2 Principle of sFIDA setup, imaging and calibration. **a** The biochemical principle of sFIDA is similar to a sandwich ELISA with capture and detection antibodies directed against overlapping epitopes of the A β N-terminus. Therefore, monomers can be captured but not detected as the epitope is already occupied. After preparation, the assay surface is imaged using dual colour fluorescence microscopy (635 and 488 nm, respectively). *Created with biorender.com* **b** Exemplary images of 500 fM SiNaPs coated with A β ₁₋₁₅, aggregates composed of 564 pg/ml A β ₁₋₄₂, a blank plasma (blank control, BC) and an AD plasma sample for the red (illumination with 635 nm) and green (illumination with 488 nm) fluorescence channels and colocalization. For imaging, the gray-scale value of 14-bit images was adjusted to min and max values of 750 and 7500, respectively. The scale bar is 50 μ m. **c** Calibration curve of 1 fM to 8 pM A β ₁₋₁₅ SiNaPs for the colocalization. **d** Dilution series of A β ₁₋₄₂ aggregates consisting of 1.1 to 18,060 pg/ml A β ₁₋₄₂ monomers. The standard deviation was calculated across the four replicates. Limit of detection (LOD) and lower limit of quantification (LLOQ) were calculated as BC with a single- or ten-fold standard deviation. Please note the logarithmic scale.

All the paragraph dedicated to HAMAS can be summarized in the last two sentences

We thank the reviewer for this comment and the proposal to shorten this paragraph. We deleted the introductory sentences and combined the results in two sentences:

HAMA: In sandwich ELISAs, heterophilic antibodies (HA) can crosslink capture and detection antibodies, causing false-positive signals²¹. By changing the capture antibody Nab228, which gave false positive signals at concentrations of 10 ng/ml or higher (Fig. 2e), to bapineuzumab (humanized equivalent to 3D6³⁴), interference from the spiked anti-mouse antibody was reduced to <0.005% at the highest concentration tested. Although a false-positive signal was observed at 1000 ng/ml HA, such concentrations are unlikely to be present in human plasma³⁵.

The paragraph “capture control” is incomprehensible to me, better explanations might help.

We added the following statement to the section “capture control” (line 230): As non-specific adherence of A β oligomers to surfaces was reported previously³⁶, we investigated the unspecific binding of the analyte to the sFIDA assay surface by introducing a control where we skip the capture antibody (capture control).

In the description of results, p.9 line 223, the correct sentence should be: “ Unexpectedly, AD and SCD subjects showed a significantly reduction plasma A β oligomer concentrations compared to controls...” not the opposite.

We adapted the wording in the manuscript (now line 246):

Unexpectedly, AD (p-value: 0.037) and SCD (p-value: 0.023) subjects showed a significantly lower plasma A β oligomer concentrations compared to control.

References

- 1 Liu, L. *et al.* An ultra-sensitive immunoassay detects and quantifies soluble A β oligomers in human plasma. *Alzheimers Dement* **18**, 1186-1202 (2022). <https://doi.org:10.1002/alz.12457>
- 2 Xia, W. *et al.* A specific enzyme-linked immunosorbent assay for measuring beta-amyloid protein oligomers in human plasma and brain tissue of patients with Alzheimer disease. *Arch Neurol* **66**, 190-199 (2009). <https://doi.org:10.1001/archneurol.2008.565>
- 3 Cline, E. N., Bicca, M. A., Viola, K. L. & Klein, W. L. The Amyloid- β Oligomer Hypothesis: Beginning of the Third Decade. *J Alzheimers Dis* **64**, S567-s610 (2018). <https://doi.org:10.3233/jad-179941>
- 4 Kurz, C., Walker, L., Rauchmann, B. S. & Perneczky, R. Dysfunction of the blood-brain barrier in Alzheimer's disease: Evidence from human studies. *Neuropathol Appl Neurobiol* **48**, e12782 (2022). <https://doi.org:10.1111/nan.12782>
- 5 An, S. S. A. *et al.* Dynamic changes of oligomeric amyloid β levels in plasma induced by spiked synthetic A β (42). *Alzheimers Res Ther* **9**, 86 (2017). <https://doi.org:10.1186/s13195-017-0310-6>

- 6 Dominguez, J. C. *et al.* Multimer Detection System-Oligomerized Amyloid Beta (MDS-OA β): A Plasma-Based Biomarker Differentiates Alzheimer's Disease from Other Etiologies of Dementia. *Int J Alzheimers Dis* **2022**, 9960832 (2022). <https://doi.org:10.1155/2022/9960832>
- 7 Santos, A. N., Simm, A., Holthoff, V. & Boehm, G. A method for the detection of amyloid-beta1-40, amyloid-beta1-42 and amyloid-beta oligomers in blood using magnetic beads in combination with Flow cytometry and its application in the diagnostics of Alzheimer's disease. *J Alzheimers Dis* **14**, 127-131 (2008). <https://doi.org:10.3233/jad-2008-14201>
- 8 Huang, Y. R. & Liu, R. T. The Toxicity and Polymorphism of β -Amyloid Oligomers. *Int J Mol Sci* **21** (2020). <https://doi.org:10.3390/ijms21124477>
- 9 Fändrich, M. Oligomeric intermediates in amyloid formation: structure determination and mechanisms of toxicity. *J Mol Biol* **421**, 427-440 (2012). <https://doi.org:10.1016/j.jmb.2012.01.006>
- 10 Lee, J. C., Kim, S. J., Hong, S. & Kim, Y. Diagnosis of Alzheimer's disease utilizing amyloid and tau as fluid biomarkers. *Experimental & Molecular Medicine* **51**, 1-10 (2019). <https://doi.org:10.1038/s12276-019-0250-2>
- 11 Babapour Mofrad, R. *et al.* Plasma amyloid- β oligomerization assay as a pre-screening test for amyloid status. *Alzheimers Res Ther* **13**, 133 (2021). <https://doi.org:10.1186/s13195-021-00873-w>
- 12 Meng, X. *et al.* Association between increased levels of amyloid- β oligomers in plasma and episodic memory loss in Alzheimer's disease. *Alzheimers Res Ther* **11**, 89 (2019). <https://doi.org:10.1186/s13195-019-0535-7>
- 13 Zhou, L. *et al.* Plasma amyloid- β oligomers level is a biomarker for Alzheimer's disease diagnosis. *Biochem Biophys Res Commun* **423**, 697-702 (2012). <https://doi.org:10.1016/j.bbrc.2012.06.017>
- 14 Shea, D. *et al.* SOBA: Development and testing of a soluble oligomer binding assay for detection of amyloidogenic toxic oligomers. *Proc Natl Acad Sci U S A* **119**, e2213157119 (2022). <https://doi.org:10.1073/pnas.2213157119>
- 15 Kulawik, A., Heise, H., Zafiu, C., Willbold, D. & Bannach, O. Advancements of the sFIDA method for oligomer-based diagnostics of neurodegenerative diseases. *FEBS Lett* **592**, 516-534 (2018). <https://doi.org:10.1002/1873-3468.12983>
- 16 Herrmann, Y. *et al.* Nanoparticle standards for immuno-based quantitation of alpha-synuclein oligomers in diagnostics of Parkinson's disease and other synucleinopathies. *Clin Chim Acta* **466**, 152-159 (2017). <https://doi.org:10.1016/j.cca.2017.01.010>
- 17 Hülsemann, M. *et al.* Biofunctionalized Silica Nanoparticles: Standards in Amyloid-beta Oligomer-Based Diagnosis of Alzheimer's Disease. *J Alzheimers Dis* **54**, 79-88 (2016). <https://doi.org:10.3233/JAD-160253>
- 18 Wang-Dietrich, L. *et al.* The amyloid-beta oligomer count in cerebrospinal fluid is a biomarker for Alzheimer's disease. *J Alzheimers Dis* **34**, 985-994 (2013). <https://doi.org:10.3233/JAD-122047>
- 19 Bloemeke, L. *et al.* Quantitative detection of alpha-Synuclein and Tau oligomers and other aggregates by digital single particle counting. *NPJ Parkinsons Dis* **8**, 68 (2022). <https://doi.org:10.1038/s41531-022-00330-x>
- 20 Hladky, S. B. & Barrand, M. A. Mechanisms of fluid movement into, through and out of the brain: evaluation of the evidence. *Fluids Barriers CNS* **11**, 26 (2014). <https://doi.org:10.1186/2045-8118-11-26>
- 21 Sehlin, D. *et al.* Interference from heterophilic antibodies in amyloid-beta oligomer ELISAs. *J Alzheimers Dis* **21**, 1295-1301 (2010). <https://doi.org:10.3233/jad-2010-100609>
- 22 Kulenkampff, K., Wolf Perez, A.-M., Sormanni, P., Habchi, J. & Vendruscolo, M. Quantifying misfolded protein oligomers as drug targets and biomarkers in Alzheimer and Parkinson diseases. *Nature Reviews Chemistry* **5**, 277-294 (2021). <https://doi.org:10.1038/s41570-021-00254-9>

- 23 Pils, M. *et al.* Development and Implementation of an Internal Quality Control Sample to Standardize Oligomer-Based Diagnostics of Alzheimer's Disease. *Diagnostics* **13**, 1702 (2023). <https://doi.org:10.3390/diagnostics13101702>
- 24 Wang, M. J. *et al.* Oligomeric forms of amyloid- β protein in plasma as a potential blood-based biomarker for Alzheimer's disease. *Alzheimers Res Ther* **9**, 98 (2017). <https://doi.org:10.1186/s13195-017-0324-0>
- 25 Kim, K. Y. *et al.* Plasma amyloid-beta oligomer is related to subjective cognitive decline and brain amyloid status. *Alzheimer's Research & Therapy* **14**, 162 (2022). <https://doi.org:10.1186/s13195-022-01104-6>
- 26 Jessen, F. *et al.* The characterisation of subjective cognitive decline. *Lancet Neurol* **19**, 271-278 (2020). [https://doi.org:10.1016/s1474-4422\(19\)30368-0](https://doi.org:10.1016/s1474-4422(19)30368-0)
- 27 Jack, C. R., Jr. *et al.* NIA-AA Research Framework: Toward a biological definition of Alzheimer's disease. *Alzheimers Dement* **14**, 535-562 (2018). <https://doi.org:10.1016/j.jalz.2018.02.018>
- 28 Jessen, F. *et al.* Design and first baseline data of the DZNE multicenter observational study on predementia Alzheimer's disease (DELCODE). *Alzheimers Res Ther* **10**, 15 (2018). <https://doi.org:10.1186/s13195-017-0314-2>
- 29 Lohmann, S. *et al.* Oral and intravenous transmission of α -synuclein fibrils to mice. *Acta Neuropathol* **138**, 515-533 (2019). <https://doi.org:10.1007/s00401-019-02037-5>
- 30 Altendorf, T. *et al.* Stabilization of Monomeric Tau Protein by All D-Enantiomeric Peptide Ligands as Therapeutic Strategy for Alzheimer's Disease and Other Tauopathies. *Int J Mol Sci* **24** (2023). <https://doi.org:10.3390/ijms24032161>
- 31 Horwitz, W. & Albert, R. The Horwitz ratio (HorRat): A useful index of method performance with respect to precision. *J AOAC Int* **89**, 1095-1109 (2006). <https://doi.org:10.1093/jaoac/89.4.1095>
- 32 Herrmann, Y. *et al.* sFIDA automation yields sub-femtomolar limit of detection for A β aggregates in body fluids. *Clin Biochem* **50**, 244-247 (2017). <https://doi.org:10.1016/j.clinbiochem.2016.11.001>
- 33 Cong, C., Zhang, W., Qian, X., Qiu, W. & Ma, C. Significant Overlap of α -Synuclein, Amyloid- β , and Phospho-Tau Pathologies in Neuropathological Diagnosis of Lewy-related Pathology: Evidence from China Human Brain Bank. *Journal of Alzheimer's Disease* **80**, 447-458 (2021). <https://doi.org:10.3233/JAD-201548>
- 34 Miles, L. A., Crespi, G. A., Doughty, L. & Parker, M. W. Bapineuzumab captures the N-terminus of the Alzheimer's disease amyloid-beta peptide in a helical conformation. *Sci Rep* **3**, 1302 (2013). <https://doi.org:10.1038/srep01302>
- 35 Mohammadi, M. M. & Bozorgi, S. Investigating the presence of human anti-mouse antibodies (HAMA) in the blood of laboratory animal care workers. *Journal of Laboratory Medicine* **43**, 87-91 (2019). <https://doi.org:doi:10.1515/labmed-2018-0084>
- 36 Willemse, E. *et al.* How to handle adsorption of cerebrospinal fluid amyloid β (1-42) in laboratory practice? Identifying problematic handlings and resolving the issue by use of the A β (42)/A β (40) ratio. *Alzheimers Dement* **13**, 885-892 (2017). <https://doi.org:10.1016/j.jalz.2017.01.010>

Reply letter

Reviewer 2, Comment 1

The paper by Blömeke et al proposed a new method for the measure A β oligomers in plasma

As pointed out by the authors the results on the relationship between the levels A β oligomers and the cognitive status found in the cohort investigated with this method are different from previous association results (decrease vs increase), the explanation of this discrepancy (total oligomers vs specific oligomers) is not convincing.

We appreciate the referee's comments regarding the discrepancies between our findings and previous results. As highlighted in the discussion, our study employed a novel method for measuring A β oligomers, specifically using different antibodies and assay conditions, which is able to explain the differing results on cognitive status associations reported in Shea et al (2022) or Wang et al (2024). Obviously, we can only speculate about pre-analytical and methodological variations that might explain the differences observed by us and others. But as long as there is no gold standard in place, all technically sound results are worth to be disseminated in the scientific community.

But the main problem of the paper is the modest differences found among the clinical groups and the up and down from SCD to MCI and AD or even control vs relative group (fig.3a) although some statistical significances are reported, the distribution of the values is substantially similar in each group.

We totally agree that the value distributions in the groups are unexpectedly large. However, they are real and this is true for the statistically significant differences between the groups, too, despite these large value distributions. Therefore, the focus of the here presented work cannot establish a test for individual diagnosis, but is rather essential to gain a comprehensive view about the oligomer development during the course of the disease and the pathophysiological role of A β oligomers in body fluids. We added the following statement to the discussion:

Line 355: "Although we demonstrate here that the sFIDA assay accurately measures total A β oligomer concentrations in plasma, the potential for using bloodborne A β oligomers as a diagnostic biomarker is limited due to the substantial overlap of individual readouts. Nevertheless, the statistically significant differences between the tested groups allow us to study the underlying pathophysiological role of A β oligomers. Owing to their central role in AD pathology, oligomers are a plausible therapeutic target to prohibit disease progression or even cure AD ⁵⁰. In pre-clinical and clinical development of anti-oligomer compounds, the quantification of plasma A β oligomers is a valuable tool to determine target engagement and to monitor therapeutic success at the molecular level."

Furthermore in Fig 3b since both group are divided in A+ and A- the comparison should be done according to this stratification (SCD A+ vs control A+). The potential contribution of these determinations in diagnosis appears modest. In terms of comprehension of oligomers production and distribution, the data in CSF would be very useful, they are mentioned as "not published yet" throughout the paper, my suggestion is to publish in a unique issue both results in CSF and plasma to give adequate information.

We appreciate the suggestion to publish both results on plasma and CSF oligomers in a unique issue. In the meantime, the results of the CSF measurements were published in *Alzheimer's and Dementia – Diagnosis, Assessment & Disease Monitoring*. Please, refer to this publication (<https://doi.org/10.1002/dad2.12589>). An increase of A β oligomer concentrations during early disease stages and a later decrease in advanced disease stages were more pronounced in CSF samples. The correlation between CSF and plasma A β oligomer concentrations, although weak in some disease stages, underline this observation.

Reference

Shea D, Colasurdo E, Smith A, Paschall C, Jayadev S, Keene CD, Galasko D, Ko A, Li G, Peskind E, Daggett V. SOBA: Development and testing of a soluble oligomer binding assay for detection of amyloidogenic toxic oligomers. *Proc Natl Acad Sci U S A*. 2022 Dec 13;119(50):e2213157119. doi: 10.1073/pnas.2213157119.

Wang SM, Kang DW, Um YH, Kim S, Lee CU, Scheltens P, Lim HK. Plasma oligomer beta-amyloid is associated with disease severity and cerebral amyloid deposition in Alzheimer's disease spectrum. *Alzheimers Res Ther*. 2024 Mar 11;16(1):55. doi: 10.1186/s13195-024-01400-3.

Blömeke L, Rehn F, Kraemer-Schulien V, Kutzsche J, Pils M, Bujnicki T, Lewczuk P, Kornhuber J, Freiesleben SD, Schneider LS, Preis L, Priller J, Spruth EJ, Altenstein S, Lohse A, Schneider A, Fliessbach K, Wiltfang J, Hansen N, Rostamzadeh A, Düzel E, Glanz W, Incesoy EI, Butryn M, Buerger K, Janowitz D, Ewers M, Pernecky R, Rauchmann BS, Teipel S, Kilimann I, Goerss D, Laske C, Munk MH, Sanzenbacher C, Spottke A, Roy-Kluth N, Heneka MT, Brosseron F, Wagner M, Wolfsgruber S, Kleineidam L, Stark M, Schmid M, Jessen F, Bannach O, Willbold D, Peters O. A β oligomers peak in early stages of Alzheimer's disease preceding tau pathology. *Alzheimers Dement (Amst)*. 2024 Apr 25;16(2):e12589. doi: 10.1002/dad2.12589. Erratum in: *Alzheimers Dement (Amst)*. 2024 May 21;16(2):e12599. PMID: 38666085; PMCID: PMC11044868.

Revision 3

Whilst we can see that you have responded to the remaining concerns of the reviewers, we notice that there are 4 points within the manuscript where you mention 'data not shown'. As all data that support the conclusions drawn must be presented in the manuscript unless they are published elsewhere, we do need to have all supporting data included.

We are keen to move towards publication of your study in Communications Medicine, but do need this additional data to be included before we can make a final decision on publication.

We therefore invite you to revise and resubmit your manuscript, to include the data mentioned. Please highlight all changes in the manuscript text file.

As pointed out by the editors, “data not shown” was mentioned 4 times in the context of three different experiments. To complete our data and allow the reader a complete view on the data, we added supplementary figures for the missing data.

1. Moreover, an upper limit of quantification (ULOQ) for SiNaPs was determined to be 256 pM, showing a 5-log dynamic range of sFIDA (data not shown)

Replaced by (line 155ff):

Moreover, an upper limit of quantification (ULOQ) for SiNaPs was determined to be 256 pM, showing a 5-log dynamic range of sFIDA (Supplementary Fig. 1).

Supplementary Fig. 1 Dynamic range of SiNaP calibration

In an independent experiment, the dynamic range of SiNaP calibration standard spiked in plasma was determined. Two-fold serial dilutions were performed starting at 512 pM down to 1 fM. ULOQ was determined at 256 pM. The average dilution linearity between 3.9 fM and 256 pM was 0.91.

2. The autofluorescence signal was below the LOD for each sample tested with a mean signal reduction of >99% compared to the signal with detection antibodies (data not shown).

Replaced by (line 220f):

The autofluorescence signal was below the LOD for each sample tested with a mean signal reduction of >99% compared to the signal with detection antibodies (Supplementary Fig. 6a).

Supplementary Figure 6: Autofluorescence of plasma samples

a Autofluorescence of 20 plasma samples was evaluated by incubation with the detection buffer without adding the capture antibody. Mean sFIDA readout was 0.61% compared to the signal with detection antibodies IC16 CF633 and Nab228 CF488 while all autofluorescence signals were below LOD. The experiment was carried out as part of the assay development, which is why there are slight deviations from the final protocol with regard to capture antibody (Nab228 at 2.5 µg/ml), blocking concentration (3% BSA) and washing steps after sample and detection antibody incubation (washing steps only with TBS).

3. In the absence of a capture antibody, SiNaPs and aggregates spiked in plasma showed a signal of 33.5% and 54.7%, respectively. However, the signals of SiNaPs and aggregates spiked in buffer were reduced by >99.9% in absence of a capture antibody (data not shown), indicating that surface binding is mediated by plasma matrix components. Similar to the calibration standards, five plasma samples tested showed non-specific binding of the analyte to the assay surface, indicated by a signal still at 77.2% (data not shown).

Replaced by (line 231-237):

In the absence of a capture antibody, SiNaPs and aggregates spiked in plasma showed a signal of 33.5% and 54.7%, respectively. However, the signals of SiNaPs and aggregates spiked in buffer were reduced by >99.9% in absence of a capture antibody (Suppl. Fig. 6b), indicating that surface binding is mediated by plasma matrix components. Similar to the calibration standards, five plasma samples tested showed non-specific binding of the analyte to the assay surface, indicated by a signal still at 77.2% (Supplementary Fig. 6c).

Supplementary Fig. 6 Capture control of SiNaPs, aggregates and plasma samples

b To test unspecific binding of the analyte to the assay surface, sFIDA assay was performed with and without capture antibody with capture control refereing to the signal without capture antibody. Leaving of the capture antibody lead to a signal reduction of >99% for 8 pM SiNaPs and A β ₁₋₄₂ aggregates (IQC-1 with 18 ng/ml A β ₁₋₄₂ monomer concentration) spiked in low cross buffer while the same concentration spiked in a plasma samples resulted in a signal reduction of 33.5% and 54.7%, respectively. **c** Signal reduction for capture control for a lower concentration of A β ₁₋₄₂ aggregates (IQC-4 with 4.45 ng/ml A β ₁₋₄₂ monomer concentration) was at 28.1% while signal reduction in capture control for plasma samples ranged from 15.5% to 30.7% with a mean signal reduction of 22.8%, respectively.

Methods (line 546-551)

Capture control: For the capture control, no capture antibody was added in the first incubation step to analyze unspecific binding of the analyte to the assay surface. All other steps were performed as described in “Assay setup”. For comparison of capture control in plasma and buffer, a cutoff of 0.01% was chosen to reduce the influence of background noise from the different matrices. Capture control refers to the signal of the analyte compared to the assay setup with capture antibody according to equation (4) and signal reduction as in equation (5).